# Associations of topic-specific peer review outcomes and institute and center award rates with funding disparities at the National Institutes of Health

**Michael S Lauer[1]\*, Jamie Doyle[2], Joy Wang[3], Deepshikha Roychowdhury[3]**

[1]Office of the Director, National Institutes of Health, Bethesda, United States; [2]Division of Clinical Innovation, National Center for Advancing Translational Sciences, Bethesda, United States; [3]Office of Extramural Research, National Institutes of Health, Bethesda, United States

**Abstract** A previous report found an association of topic choice with race-based funding disparities among R01 applications submitted to the National Institutes of Health ('NIH') between 2011 and 2015. Applications submitted by African American or Black ('AAB') Principal Investigators ('PIs') skewed toward a small number of topics that were less likely to be funded (or 'awarded'). It was suggested that lower award rates may be related to topic-related biases of peer reviewers. However, the report did not account for differential funding ecologies among NIH Institutes and Centers ('ICs'). In a re-analysis, we find that 10% of 148 topics account for 50% of applications submitted by AAB PIs. These applications on 'AAB Preferred' topics were funded at lower rates, but peer review outcomes were similar. The lower rate of funding for these topics was primarily due to their assignment to ICs with lower award rates, not to peer-reviewer preferences.

**\*For correspondence:**
Michael.Lauer@nih.gov

**Competing interests:** The authors declare that no competing interests exist.

## Introduction

Data recently reported by *Hoppe et al., 2019* from the National Institutes of Health ('NIH') suggest that part of the well-documented funding disparity (*Ginther et al., 2011*) affecting African-American Black ('AAB') principal investigators ('PIs') may be related to the topic of their applications. The authors of that report (including the first author of this report) found that topic choice accounted for over 20% of the disparity and wondered whether biases on the part of peer reviewers might explain why some application topics fare less well when submitted to the NIH for consideration of funding. In other words, peer reviewers might prefer certain topics over others, and specifically may be biased against applications focused on topics preferred by AAB PI's.

However, peer review outcomes are not the only determinant of funding. Across the agency, grants are not simply awarded in order of peer review scores (*Taffe and Gilpin, 2021*). One reason is that applications submitted to the NIH are not just submitted to NIH; they are assigned to one of 24 grant-issuing institutes or centers ('ICs') that in turn make decisions about which applications to fund. The proportion of applications funded (or 'award rate') varies across ICs; therefore, we can think of the NIH process as not one competition hinging entirely on peer review but rather as 24 separate competitions. The variability of award rates relates to differences in number of applications each IC receives, available funds, and IC priorities.

*Hoppe et al., 2019* did not account for IC assignment or variation in IC-specific award rates. It is possible that the apparent link between topic choice and funding disparities may reflect differences in IC assignment, since ICs receive applications according to alignment with their stated mission. For example, applications focusing on cancer epidemiology are more likely to be assigned to the

National Cancer Institute while those focusing on basic human biology are more likely to be assigned to the National Institute of General Medical Sciences. If award rates at the National Institutes of General Medical Sciences are higher than at the National Cancer Institute, it might appear that NIH 'prefers' basic human biology over cancer epidemiology. While the former topic does fare better with a higher likelihood of funding, this may be largely because of different IC award rates as opposed to differences in how the topics are received by peer reviewers.

We therefore re-analyzed the data from *Hoppe et al., 2019* focusing on topic-specific peer review outcomes according to AAB PI preferences and on the possible role of IC assignment in application outcomes. To minimize well-documented biases related to repeated looks by the peer review system on individual proposals (from resubmissions [*Lauer, 2017*] or competing renewals [*Lauer, 2016*]) we focus on de novo applications submitted to the NIH for the first time.

## Materials and methods

These analyses are based on R01 applications submitted to NIH between 2011 and 2015. *Hoppe et al., 2019* described in detail NIH peer review processes and the 'Word2vec' algorithm *Mikolov et al., 2013* used to designate a topic for each application. Briefly, each application is assigned to a peer review group; approximately 2% of applications are withdrawn prior to review for administrative reasons. After a preliminary pre-meeting review, approximately half are deemed to be potentially meritorious and are therefore discussed during a formally convened meeting. After the meeting, each discussed application receives a priority score ranging from 10 (best) to 90 (worst); many, but not all, applications also receive a 'percentile ranking' to account for differences in how individual peer review groups calibrate their scores.

Applications are not only assigned to peer review groups; they are also assigned to ICs. The Center of Scientific Review (CSR) Division of Receipt and Referral makes assignments (*CSR, 2021*) based on IC interests, specific Funding Opportunity Announcements (as appropriate), and sometimes on applicant request. Some applications receive a dual assignment (that is, two ICs with one primary and one secondary), but changes in primary IC assignment occur less than 2% of the time. ICs ultimately make decisions about which applications to fund, with funding decisions based on peer review scores, strategic priorities, and availability of funds (*NIH, 2021c*).

### Data and descriptive measures

To eliminate biases due to prior reviews and to maximize the likelihood that the analysis data frame reflects comparable observations with independently distributed variables, we focused on applications coming to the NIH for the first time; in other words, we excluded resubmissions (*Lauer, 2017*) and competing renewals (*Lauer, 2016*) which are known to systematically fare better on peer review. Each application was designated to one of 148 topics identified by the Word2vec algorithm (*Mikolov et al., 2013*). *Hoppe et al., 2019* stated there were 150 topics, but subsequently discovered data processing errors for two topics yielded 148 for the re-analysis.

For each topic, we counted the number of applications of submitted by AAB PIs. We assessed AAB preference for each topic *t* by calculating the proportion of all AAB applications within that topic, that is

$$P_{AAB} = (\sum_{i=1}^{n} aab)/(\sum_{i=1}^{N} aab) \tag{1}$$

where *i* refers to individual applications, *n* is the number of applications in topic *t*, *aab* is an indicator of whether application *i* had an African American or Black contact PI (1 if yes, 0 if no), and *N* is total number of applications (across all topics). For each IC, we calculated award rates *A* as number of applications funded divided by number of applications assigned, that is

$$A_{IC} = \frac{\sum_{i=1}^{m} a}{m} \tag{2}$$

where *m* is the number of applications received by the IC and *a* is an indicator of whether an application *i* was awarded (1 if yes, 0 if no). Similarly, for each topic we calculated awards rates as

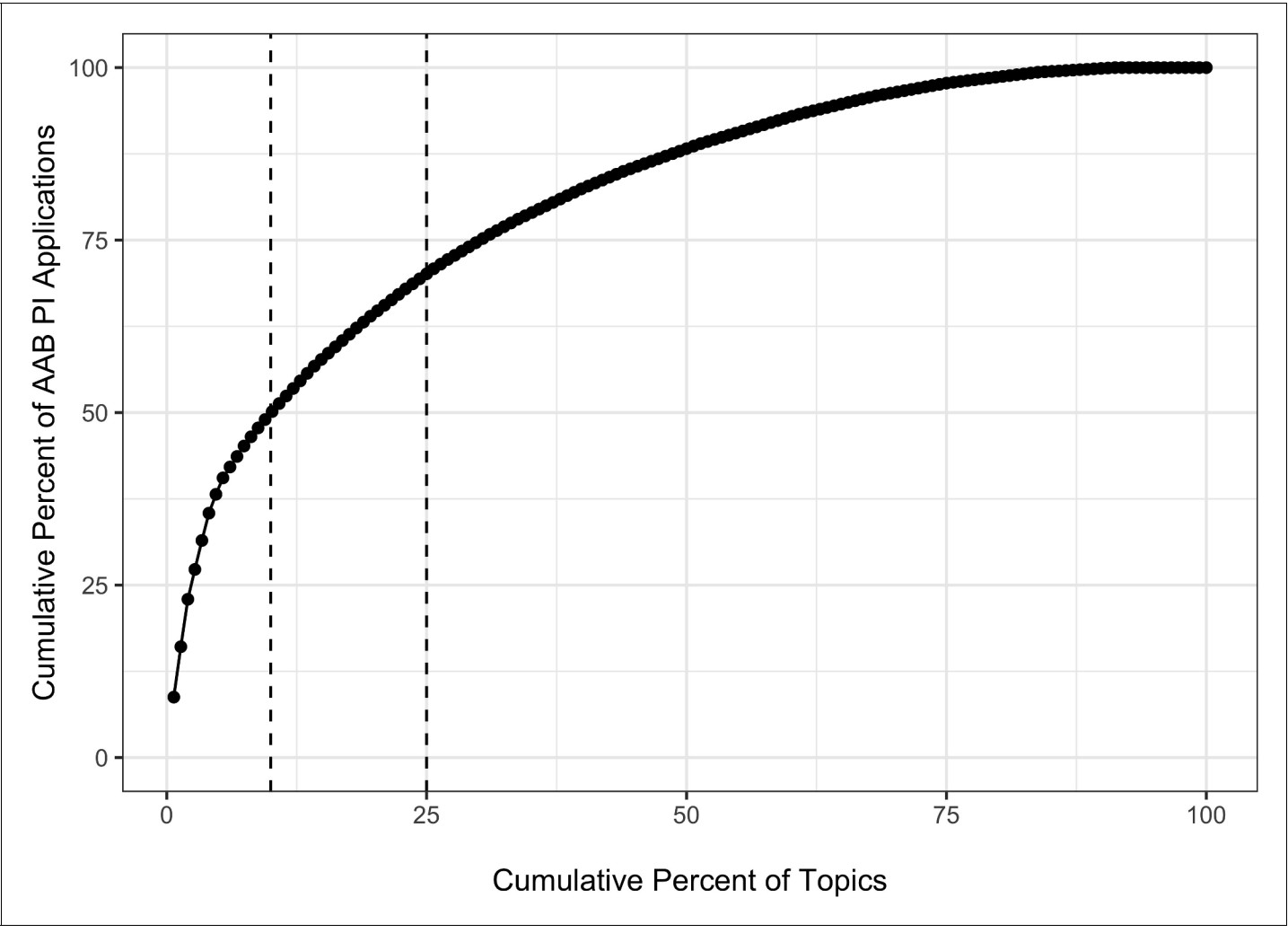

**Figure 1.** Cumulative percentage plot showing the association of cumulative percentage of applications submitted by AAB Principal Investigators and cumulative percentage of topics. Each dot represents a topic; the first dot on the lower left corner shows that this one topic accounted for over 8% of all applications submitted by an AAB PI. Dashed vertical lines show that 10% of the topics accounted for 50% of the applications submitted by AAB Principal Investigators and that 25% of the topics accounted for 70% of the applications submitted by AAB Principal Investigators. AAB = African American or Black.

$$A_t = \frac{\sum_{i=1}^{n} a}{n} \qquad (3)$$

where again $a$ is an indicator of whether an application $i$ was awarded (1 if yes, 0 if no) and $n$ is the number of applications in the topic. We estimated a predicted award rate for each topic by designating the probability for funding for each application $i$ as its IC award rate (in other words, if an application $i$ was assigned to NEI, the probability of funding was 0.15). Thus, the predicted topic award rate was

$$\hat{A}_t = \frac{\sum_{i=1}^{n} \hat{a}}{n} \qquad (4)$$

where $\hat{a}$ is the award rate for the IC to which each application $i$ is assigned and $n$ is the number of applications in the topic.

For each topic, we calculated two peer review outcomes. The proportion of applications making it to discussion (or 'discussed') was

**Table 1.** Application characteristics according to Institute or Center (IC), as ranked by award rate (*Equation 2*, highest to lowest). AAB = African American or Black; PI = Principal Investigator; EY = National Eye Institute; DC = National Institute of Deafness and Communications Disorders; GM = National Institute of General Medical Sciences; DE = National Institute of Dental and Craniofacial Research; MH = National Institute of Mental Health; DA = National Institute on Drug Abuse; NS = National Institute of Neurological Disorders and Stroke; NR = National Institute of Nursing Research; HL = National Heart, Lung, and Blood Institute; AI = National Institute of Allergy and Infectious Diseases; ES = National Institute of Environmental Health Sciences; DK = National Institute of Diabetes and Digestive and Kidney Disease; AA = National Institute on Alcohol Abuse and Alcoholism; AG = National Institute on Aging; EB = National Institute of Biomedical Imaging and Bioengineering; CA = National Cancer Institute; HD = Eunice Kennedy Shriver National Institute of Child Health and Human Development; MD = National Institute on Minority Health and Health Disparities; AR = National Institute of Arthritis and Musculoskeletal and Skin Diseases. Data for ICs with cell sizes not exceeding 11 are not shown due to privacy concerns. Final columns shows 2015 appropriations in billions of dollars.

| IC | Applications(N) | Awards(N) | Award rate(%) | AAB PI | AAB(%) | Appropriations($B) |
|----|----|----|----|----|----|----|
| EY | 2026 | 315 | 15.5 | 14 | 0.69 | 0.684 |
| GM | 9781 | 1266 | 12.9 | 113 | 1.16 | 2.371 |
| DC | 1145 | 141 | 12.3 | 13 | 1.14 | 0.405 |
| DE | 1282 | 140 | 10.9 | 26 | 2.03 | 0.400 |
| MH | 5067 | 536 | 10.6 | 87 | 1.72 | 1.463 |
| DA | 3606 | 359 | 10.0 | 66 | 1.83 | 1.029 |
| NS | 6786 | 668 | 9.8 | 77 | 1.13 | 1.605 |
| NR | 1180 | 107 | 9.1 | 55 | 4.66 | 0.141 |
| HL | 11,325 | 995 | 8.8 | 171 | 1.51 | 2.998 |
| ES | 2193 | 188 | 8.6 | 44 | 2.01 | 0.745 |
| AI | 9421 | 791 | 8.4 | 196 | 2.08 | 4.359 |
| DK | 6986 | 538 | 7.7 | 110 | 1.57 | 1.900 |
| AA | 1423 | 105 | 7.4 | 16 | 1.12 | 0.447 |
| EB | 1777 | 131 | 7.4 | 15 | 0.84 | 0.330 |
| AG | 4211 | 302 | 7.2 | 52 | 1.23 | 1.199 |
| CA | 15,906 | 1045 | 6.6 | 206 | 1.30 | 4.950 |
| HD | 5454 | 340 | 6.2 | 168 | 3.08 | 1.287 |
| MD | 829 | 51 | 6.2 | 123 | 14.84 | 0.269 |
| AR | 2656 | 161 | 6.1 | 30 | 1.13 | 0.522 |

$$D_t = \frac{\sum_{i=1}^{n} d}{n} \tag{5}$$

where $d$ is an indicator of whether an application $i$ was discussed (1 if yes, 0 if no) and $n$ is the number of applications in the topic. For those applications that were discussed and therefore received a priority score, we calculated a topic-specific mean score as

$$\overline{s}_t = \frac{\sum_{i=1}^{l} s}{l} \tag{6}$$

where $l$ is the number of applications in that topic that made it to discussion and $s$ is the priority score of each application $i$.

## Regression analyses

*Figure 1* shows the association of cumulative percentage of AAB applications by the cumulative percentage of topics. Consistent with *Hoppe et al., 2019*, there was a nonrandom distribution whereby 10% of topics (or 15 topics) accounted for 50% of AAB applications, while 25% of the topics accounted for 70% of AAB applications. We designed a topic as 'AAB Preferred' if it was among the 15 topics that accounted for 50% of AAB applications.

**Table 2.** Application review and funding outcomes according to whether Institute or Center received a higher or lower proportion of applications from AAB principal investigators. AAB = African American or Black; PI = Principal Investigator.

| Characteristic or outcome | | Higher AAB | Lower AAB |
|---|---|---|---|
| Total N (%) | | 18262 (18.9) | 78435 (81.1) |
| PI AAB | Yes | 579 (3.2) | 1064 (1.4) |
| Discussed | Yes | 8809 (48.2) | 35845 (45.7) |
| Priority Score | Median (IQR) | 39.0 (30.0 to 48.0) | 40.0 (30.0 to 48.0) |
| Percentile Ranking | Median (IQR) | 34.0 (21.0 to 45.0) | 33.0 (21.0 to 44.0) |
| Awarded | Yes | 1440 (7.9) | 6941 (8.8) |

To further assess the association of topic choice with peer review outcomes, we focused on applications that were discussed and therefore received a priority score. We constructed a plot of topic-specific mean peer review scores according to number of applications in each topic. We expected to find a greater variance of mean scores for topics receiving fewer applications ('regression to the mean'). We generated a linear regression model to estimate a predicted mean score for each topic based on topic size (namely the number of submitted applications that were discussed for each topic), and calculated a residual for each topic by subtracting from each topic-specific mean score the model-based predicted mean score. We compared the distribution of residuals for AAB preferred topics and other topics.

We next performed a series of probit regression analyses of application funding as

$$Y_i = \beta_0 + \beta_1 PI_{AAB} + \beta_2 T_{AABP} + \beta_3 A_{IC} + \beta X + \epsilon \tag{7}$$

where $PI_{AAB}$ is an indicator variable for whether application $i$ had an African American or Black PI (yes or no), $T_{AABP}$ is an indicator variable for whether application $i$ focused on an AAB Preferred topic (yes or no), $A_{IC}$ is the IC award rate (*Equation 2*), and $X$ corresponds to variables including 'Early Stage Investigator' status, whether application $i$ designated more than one PI, and whether the proposed research in application $i$ involved animal and/or human subjects. Early Stage Investigators were those PIs within 10 years of their terminal research degree or completion of clinical training. Akaike Information Criteria, Bayesian Information Criteria, and Log Likelihood values informed model strength.

All analyses used R (*R Foundation, 2021*) packages, including tidyverse (*Wickham et al., 2019*), ggplot2 (*Wickham, 2016*), finalfit (*Harrison, 2020*), and texreg (*Leifeld, 2013*).

## Results

Of 157,405 applications received, there were, after exclusion of pre-review withdrawals, resubmissions and competing renewals, 96,697 applications considered and reviewed by NIH for the first time. Of these 8381 were funded, for an overall award rate of 9%. The vast majority of applications (88%) were reviewed in the NIH Center for Scientific Review (or CSR). There were 1643 applications, or 2%, submitted by AAB PIs. *Table 1* shows IC-specific values for applications received, applications funded, award rates (*Equation 2*), and percent applications coming from AAB PIs. Of note, award rates varied from 6% to 15%, while the proportion of applications with AAB PIs ranged from <1% to nearly 15%. We also show 2015 appropriations for each IC (*NIH, 2021b*).

*Table 2* shows review and funding outcomes for applications according to whether the assignment was to an IC in the top quartile of proportion of applications with AAB PIs ('Higher AAB'). These ICs were the National Institute of Allergy and Infectious Diseases, the National Institute of Dental and Craniofacial Research, the National Institute of Child Health and Human Development, the National Institute of Minority Health and Disparities, the National Institute of Nursing Research, and the Fogarty International Center. Applications assigned to Higher AAB ICs were three times more likely to come from AAB PIs. Review outcomes – proportion discussed and, for those applications that were discussed at peer review meetings, priority scores and percentile rankings – were

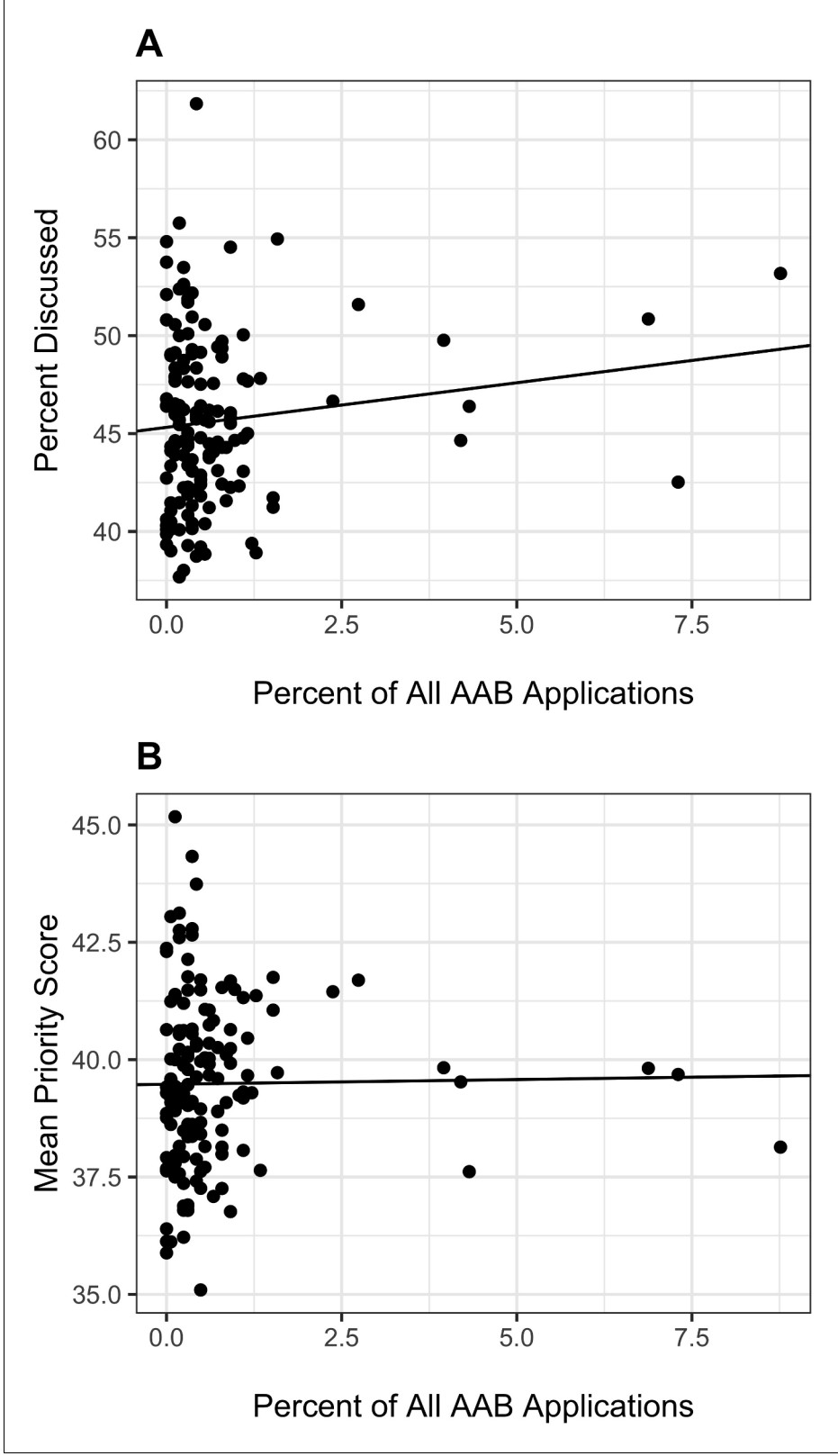

**Figure 2.** Topic peer review outcomes according to proportion of AAB applications linked to specific topics. AAB = African American or Black. Panel A: Scatter plot of topic-specific probability of discussion according to proportion of AAB applications. Each dot refers to a topic; the three topics furthest to the right on the X-axis correspond to the three topics at the lower left of *Figure 1*. The line is based on a linear regression of proportion of topic-specifc applications discussed on proportion of AAB applications. The slope of the line was not significant (p=0.12). Panel **B**: Scatter plot of topic-

*Figure 2 continued on next page*

*Figure 2 continued*

specific mean priority score according to proportion of AAB applications. Lower mean priority scores correspond to better, more favorable reviews. Each dot refers to a topic; the three topics furthest to the right on the X-axis correspond to the three topics at the lower left of *Figure 1*. The line is based on a linear regression of proportion of topic-specific mean priority score on proportion of AAB applications; analyses are based on those applications that were discussed. The slope of the line was not significant (p=0.87).

similar in both groups. Despite the similar review outcomes, these applications were 10% less likely to be funded. These descriptive data show that two IC-defined cohorts of applications with similar review outcomes can have substantively different funding outcomes.

## Review outcomes according to topic

*Figure 2* shows the topic-based peer review outcomes according to the proportion of total AAB applications linked to each topic. There was a non-significant trend whereby applications focusing on topics that accounted for a greater proportion of AAB applications (*Equation 1*) were more likely to make it to discussion (*Equation 5*)(Panel A, p=0.12). When we focused on applications that were discussed and therefore received a priority score, there was no association of mean score for each topic (*Equation 6*) with proportion of total AAB applications linked to each topic (*Equation 1*) (Panel B, p=0.87).

To gain greater insights into possible peer review biases against topics preferred by AAB PIs, *Figure 3*, Panel A, shows the mean priority score by topic (*Equation 6*; again, only discussed applications receive priority scores) according to the topic size, namely the number of submitted applications that were discussed for each topic. As would be expected topics of smaller size showed greater variability, a manifestation of regression to the mean.

The line in *Figure 3*, Panel A, is based on a linear regression of mean score according to topic size. Although the slope was slightly negative (coeffecient −0.0002634), the association was not significant (p=0.63). Among AAB preferred topics (orange dots), there were 5 more than 1 point above the line (meaning with scores worse than predicted), while there were 3 more than 1 point below the line (meaning with scores better than predicted). The remaining seven topics had mean scores that were within 1 point of the predicted value.

For each topic, we calculated a residual by subtracting from the topic-specific mean priority score (*Equation 6*) the predicted mean priority score; we weighted the residuals by the topic size, as the larger topics contribute more information. *Figure 3*, Panel B, shows the distribution of the weighted residuals according to topic type. Residuals were more positive (i.e. worse) for AAB preferred topics. However, the absolute differences were small, much less than one priority score point (over a possible range of 10–90, with topic-specific mean values ranging from 35 to 45).

## Funding outcomes according to topic: descriptive findings

*Table 3* shows peer review and funding outcomes according to whether applications were focused on the 15 (or 10% of) topics that made up 50% of all applications with AAB PIs ('AAB Preferred' topics). Consistent with the absence of association with proportion of AAB applications linked to each topic (*Figure 2*), peer review outcomes were similar in the two groups. However, applications focusing on AAB Preferred topics were 7% less likely to be funded. *Table 4* shows similar findings when focusing on the 37 (or 25% of) topics that made up 70% of all applications with AAB PIs.

Why do applications on AAB Preferred topics have worse funding outcomes despite similar peer review assessments? *Table 3* shows that applications on AAB Preferred topics (those 10% of topics that accounted for 50% of all AAB applications) were 42% more likely to be assigned to Higher AAB ICs. The scatter plot in *Figure 4* shows IC award rate according to the proportion of applications assigned to it that focus on AAB Preferred topics. ICs with receiving a higher percentage of AAB Preferred topic applications tended to have lower award rates (r = −0.35, p=0.08).

The association between IC award rates and application success is demonstrated in *Figure 5*, Panel A, which shows that observed topic-specific award rates (*Equation 3*) were strongly correlated with predicted award rates (*Equation 4*), that is award rates that would be expected based solely on IC-specific award rates (r = 0.62, p<0.0001). *Figure 5*, Panel B shows a tendency toward lower predicted topic-specific award rates (*Equation 4*) with increasing proportion of AAB applications (*Equation 1*) linked to each topic (p=0.06). Panel C shows that the actual topic-specific award rates

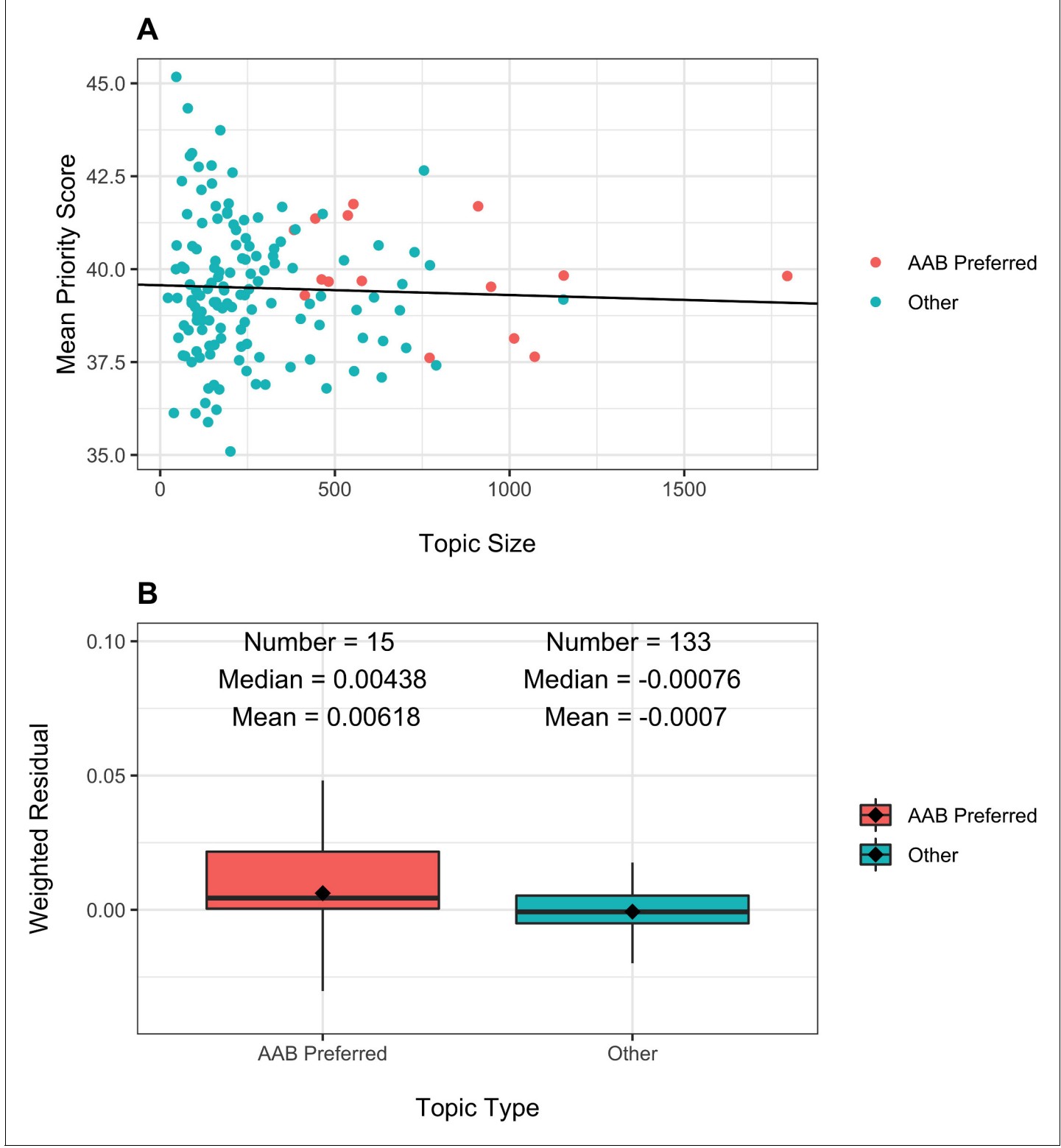

**Figure 3.** Topic peer review scores according to number of applications received ('Topic Size') and topic type (AAB Preferred or Other). Panel **A**: Scatter plot of topic-specific mean peer review scores according to topic size. Each dot refers to a topic, with orange dots AAB preferred topics and green dots all others. The line is based on a linear regression of mean peer review scores on topic size. The slope of the line was not significant (p=0.63). Panel **B**: Distribution of weighted residuals of topic-specific mean review scores. Residuals are calculated as the distance between the dots in Panel A and the regression line, and are then weighted by topic size.

**Table 3.** Application review and funding outcomes according to whether topic was among those that accounted for half of all AAB applications.
Abbreviations as in *Table 2*.

| Characteristic or outcome | | AAB preferred | Other |
|---|---|---|---|
| Total N (%) | | 24416 (25.3) | 72281 (74.7) |
| PI AAB | Yes | 824 (3.4) | 819 (1.1) |
| Discussed | Yes | 11515 (47.2) | 33139 (45.8) |
| Priority Score | Median (IQR) | 40.0 (31.0 to 48.0) | 39.0 (30.0 to 48.0) |
| Percentile Ranking | Median (IQR) | 34.0 (22.0 to 45.0) | 33.0 (20.0 to 44.0) |
| Assigned IC AAB Proportion | Higher AAB | 5925 (24.3) | 12337 (17.1) |
| Awarded | Yes | 1992 (8.2) | 6389 (8.8) |

(*Equation 3*) were non-significantly negatively associated with increasing proportion of AAB applications (*Equation 1*) (p=0.76).

## Funding outcomes according to topic: probit regression models

*Table 5* shows the association of an application with an AAB PI with the probability of funding. Consistent with *Hoppe et al., 2019* and prior literature (*Ginther et al., 2011*), AAB PI applications had a lower likelihood of funding (Model 1, negative regression coefficient for AAB Principal Investigator, p<0.001). Adjusting for the topic (AAB Preferred or Other) reduced the absolute value of the regression coefficient for race by 5% (Model 2); similarly adjusting for IC assignment (Higher or Lower AAB) reduced the absolute value of the regression coefficient by 5% (Model 3). However, adjusting for the award rate of the assigned IC reduced the absolute value of the regression coefficient for race by 14% (Model 4).

*Table 6* focuses on topic and funding outcomes. Without consideration of other variables, an AAB preferred topic was associated with a lower probability of funding (Model 1, negative regression coefficient for AAB preferred topic, p<0.01). However, after adjusting for IC award rate alone (Model 2, regression coefficient for AAB preferred topic close to zero, p=NS) as well as IC award rate and other characteristics (whether the PI is an early stage investigator, whether the application included more than one PI, and whether the proposed research included human and/or animal subjects), there was no association between AAB preferred topics and funding (Model 3, regression coefficient for AAB preferred topic close to zero, p=NS). The IC award rate was strongly associated with likelihood of funding (Models 2 and 3, regression coefficient positive, p<0.001).

## Resubmission and single-PI applications

We repeated the main analyses, but this time focusing on resubmission applications, which as a general rule are assigned to the same IC as the original submission. The findings were similar, except

**Table 4.** Application review and funding outcomes according to whether topic was among those that accounted for 70% of all AAB applications.
Abbreviations as in *Table 2*.

| Characteristic or outcome | | AAB preferred | Other |
|---|---|---|---|
| Total N (%) | | 46108 (47.7) | 50589 (52.3) |
| PI AAB | Yes | 1152 (2.5) | 491 (1.0) |
| Discussed | Yes | 21485 (46.6) | 23169 (45.8) |
| Priority Score | Median (IQR) | 40.0 (31.0 to 48.0) | 39.0 (30.0 to 48.0) |
| Percentile Ranking | Median (IQR) | 34.0 (21.0 to 45.0) | 33.0 (20.0 to 44.0) |
| Assigned IC AAB Proportion | Higher AAB | 11424 (24.8) | 6838 (13.5) |
| Awarded | Yes | 3852 (8.4) | 4529 (9.0) |

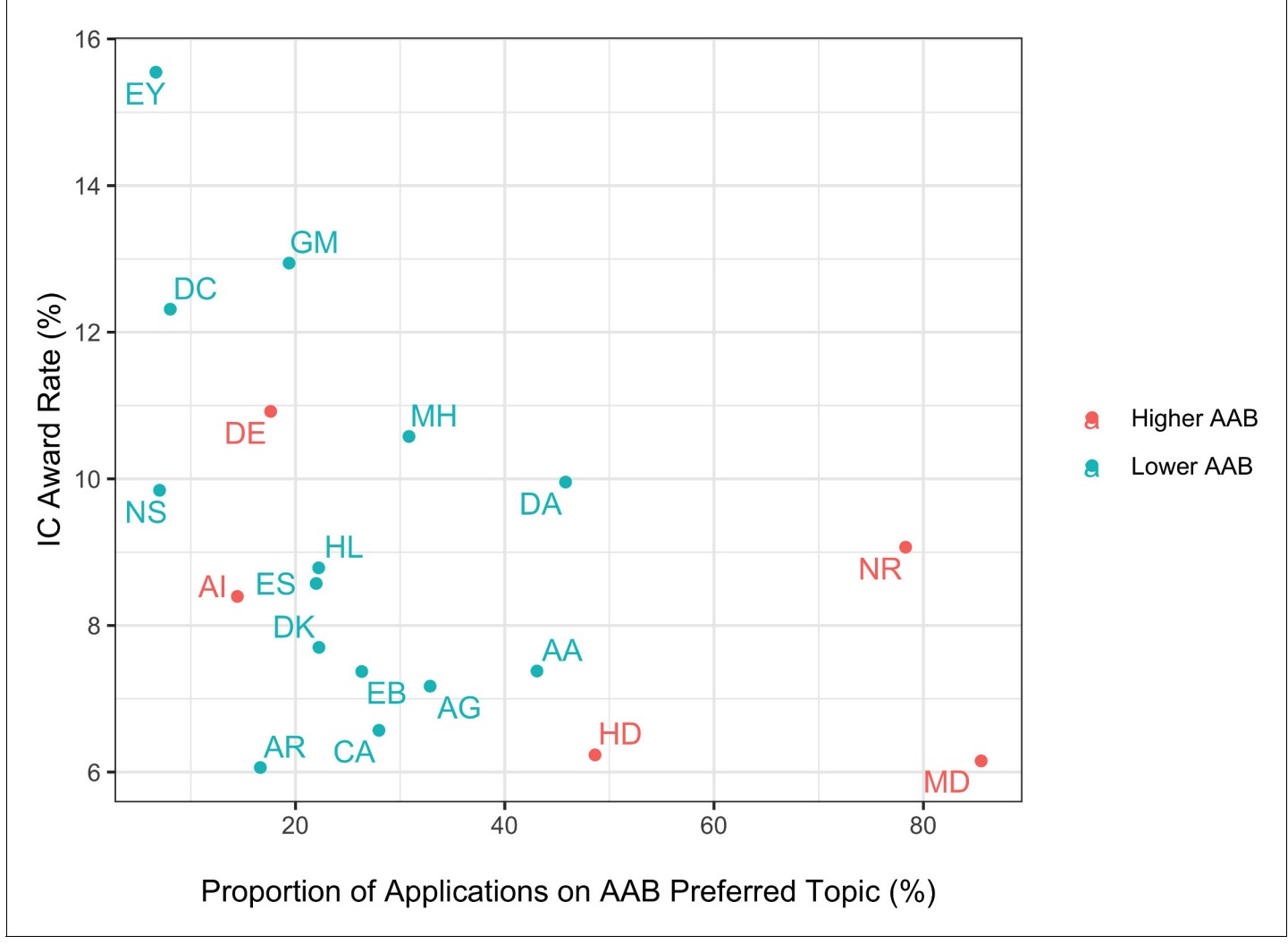

**Figure 4.** Scatter plot of IC-specific award rates according to proportion of IC applications that focus on AAB Preferred topics. In other words, the X-axis value would be 33% if one-third of all IC applications focused on AAB Preferred topics, namely those 15 topics that together accounted for 50% of all applications with AAB PIs. The orange-colored ICs are those in which the proportion of applications from AAB PIs were in the top quartile. AAB = African American or Black; IC = Institute or Center; EY = National Eye Institute; DC = National Institute of Deafness and Communications Disorders; GM = National Institute of General Medical Sciences; DE = National Institute of Dental and Craniofacial Research; MH = National Institute of Mental Health; DA = National Institute on Drug Abuse; NS = National Institute of Neurological Disorders and Stroke; NR = National Institute of Nursing Research; HL = National Heart, Lung, and Blood Institute; AI = National Institute of Allergy and Infectious Diseases; ES = National Institute of Environmental Health Sciences; DK = National Institute of Diabetes and Digestive and Kidney Disease; AA = National Institute on Alcohol Abuse and Alcoholism; AG = National Institute on Aging; EB = National Institute of Biomedical Imaging and Bioengineering; CA = National Cancer Institute; HD = Eunice Kennedy Shriver National Institute of Child Health and Human Development; MD = National Institute on Minority Health and Health Disparities; AR = National Institute of Arthritis and Musculoskeletal and Skin Diseases. ICs that receive relatively more applications on AAB Preferred topics tended to have lower award rates (r = −0.35, p=0.08). Data for ICs with cell sizes not exceeding 11 are not shown due to privacy concerns.

that, as expected, the absolute award rates were higher. We also conducted a separate series of analyses which repeated our primarily analyses but focusing solely on single-PI applications. Again, findings were similar (see Appendix 1 and Appendix 2).

## Discussion

Among over 96,000 R01 applications submitted to NIH between 2011 and 2015, 2% were submitted by AAB PIs. Their applications were skewed towards a relatively small group of 'AAB Preferred' topics (*Figure 1*); 10% of 148 topics accounted for 50% of AAB applications. Applications on AAB

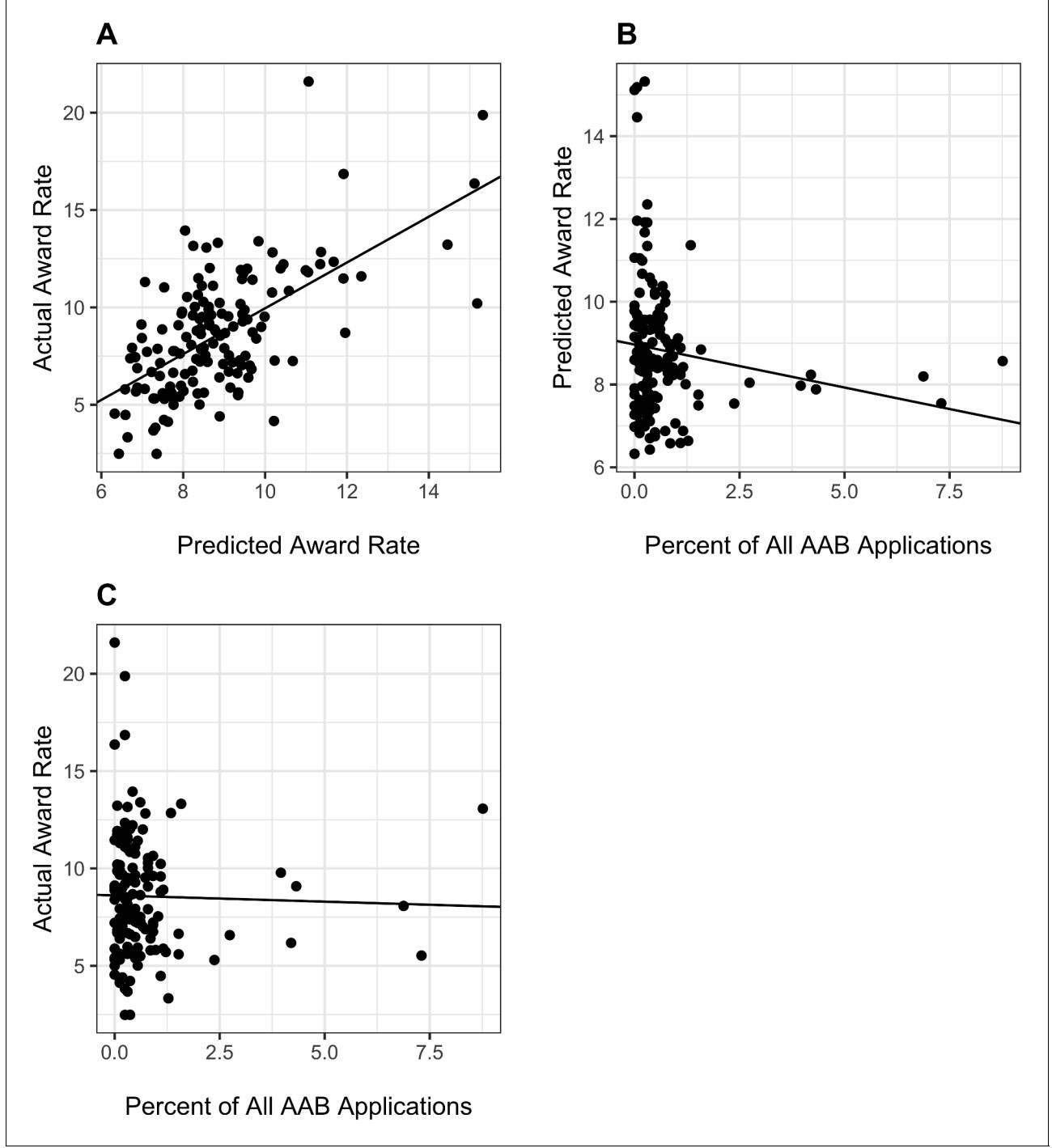

**Figure 5.** Descriptive analyses of actual (*Equation 3*) and predicted (*Equation 4*) topic-specific award rates.  AAB = African American or Black. Panel **A**: Association of actual vs predicted award rates. Each dot refers to a topic. The line is based on a linear regression of actual award rate on predicted award rate (r = 0.62, p<0.0001). Panel **B**: Association of predicted award rate (*Equation 4*) with proportion of AAB applications (*Equation 1*). Each dot refers to a topic; the three topics furthest to the right on the X-axis correspond to the 3 topics at the lower left of *Figure 1*. The line is based on a linear regression of predicted award rate on percent of all AAB applications; the negative slope was borderline signifiant (p=0.06). Panel **C**: Association of actual award rate (*Equation 3*) with proportion of AAB applications (*Equation 1*). Each dot refers to a topic; the three topics furthest to the right on the X-axis correspond to the three topics at the lower left of *Figure 1*. The line is based on a linear regression of actual award rate on percent of all AAB applications; the slope was non-signficant (p=0.76).

**Table 5.** Probit Regression Models (regression coefficients and standard errors) with focus on the PI. Model 1 shows the univariable association of funding success according to whether the PI is AAB. The negative non-zero cofficient indicates that applications with AAB PIs are less likely to be funded; the p-value refers to the likelihood that the difference from zero would be due to chance were the null hypothesis of no association to be true. Model 2 adjusts for topic, Model 3 adjusts for IC assignment, and Model 4 adjusts for IC award rate. Note that the absolute value for the regression coefficient linking AAB PI to funding outcome decreases with each of these adjustments, with the greatest reduction after adjusting for IC award rate. AIC = Akaike Information Criterion; BIC = Bayesian Information Criterion; Num. obs. = Number of Observations. Other abbreviations same as in *Tables 1* and *2*.

| | Model 1 | Model 2 | Model 3 | Model 4 |
|---|---|---|---|---|
| Intercept | −1.359*** | −1.349*** | −1.348*** | −1.903*** |
| | (0.006) | (0.007) | (0.006) | (0.020) |
| AAB Principal Investigator | −0.184*** | −0.174*** | −0.174*** | −0.158** |
| | (0.049) | (0.049) | (0.049) | (0.050) |
| AAB Preferred Topic | | −0.040** | | |
| | | (0.013) | | |
| IC Higher AAB Applications | | | −0.060*** | |
| | | | (0.015) | |
| IC Award Rate | | | | 0.061*** |
| | | | | (0.002) |
| AIC | 56996.279 | 56989.156 | 56982.092 | 56175.409 |
| BIC | 57015.238 | 57017.594 | 57010.530 | 56203.847 |
| Log Likelihood | −28496.140 | −28491.578 | −28488.046 | −28084.705 |
| Deviance | 56992.279 | 56983.156 | 56976.092 | 56169.409 |
| Num. obs. | 96697 | 96697 | 96697 | 96697 |

***$p<0.001$; **$p<0.01$; *$p<0.05$.

Preferred topics had similar review outcomes as those on other topics (*Figure 2* and *Table 3*) but were less likely to be funded. The lower award rates for applications on AAB Preferred applications (*Table 5*, Model 1) were no longer seen after accounting for IC award rates (*Table 6*, Models 2 and 3).

These observations reflect that there are two factors at play in determining whether an application submitted to NIH will be funded. The first, well known to all involved with NIH system, is peer review; those applications that receive better scores are more likely to be funded. But there is a second factor, namely the funding ecology of the IC to which the application is assigned. As shown in *Table 2* applications with similar peer review outcomes are less likely to be funded if they are assigned to ICs with lower overall award rates. AAB PIs are more likely to submit applications to ICs with lower award rates, and applications (whether submitted by AAB or other PIs) that focus on AAB Preferred topics tended to be assigned to ICs with lower award rates (*Figure 4*).

*Hoppe et al., 2019* found that topic choice accounted for over 20% of funding disparities that adversely effect AAB PIs. We confirm this, but find that IC assignment (which, of course, is linked to topic) explains the disparities just as well, and that IC award rates explain the disparities even better (*Table 5*, 14% vs 5% reduction in absolute value of regression coefficient for AAB PI). Furthermore, after accounting for IC award rate, we found no association between topic choice and funding outcome (*Table 6*).

There is variability in how well different topics fare at peer review, but inspection of *Figure 3* suggests that much of this variability reflects instability of estimates stemming from smaller sample sizes. Many topics that are not favored by AAB PIs receive better (lower) priority scores than the overall average, but many other such topics receive worse scores (*Figure 3*, Panel A). An inspection of weighted residuals suggest that AAB Preferred topics may fare a bit worse (*Figure 3*, Panel B), but

**Table 6.** Probit Regression Models (regression coefficients and standard errors) with focus on topic type.

Model 1 shows the univariable association of funding success according to whether the topic is AAB preferred. Model 2 shows results according to topic choice and IC award rate. Model 3 includes early stage investigator status, whether applications had multiple PIs, and whether the application included research on human subjects and/or animal subjects. AIC = Akaike Information Criterion; BIC = Bayesian Information Criterion; Num. obs. = Number of Observations. Other abbreviations same as in *Tables 1* and *2*.

|  | Model 1 | Model 2 | Model 3 |
|---|---|---|---|
| Intercept | −1.351*** | −1.904*** | −1.869*** |
|  | (0.007) | (0.021) | (0.027) |
| AAB Preferred Topic | −0.044** | −0.005 | −0.013 |
|  | (0.013) | (0.014) | (0.015) |
| IC Award Rate |  | 0.061*** | 0.057*** |
|  |  | (0.002) | (0.002) |
| AAB Principal Investigator |  |  | −0.170*** |
|  |  |  | (0.050) |
| Early Stage Investigator |  |  | 0.149*** |
|  |  |  | (0.015) |
| Multi-PI Application |  |  | 0.099*** |
|  |  |  | (0.015) |
| Human Subjects |  |  | −0.056 |
|  |  |  | (0.014) |
| Animal Subjects |  |  | −0.052*** |
|  |  |  | (0.014) |
| AIC | 57000.243 | 56185.926 | 56047.521 |
| BIC | 57019.201 | 56214.364 | 56123.355 |
| Log Likelihood | −28498.121 | −28089.963 | −28015.760 |
| Deviance | 56996.243 | 56179.926 | 56031.521 |
| Num. obs. | 96697 | 96697 | 96697 |

***$p<0.001$; **$p<0.01$; *$p<0.05$.

to a lower degree than the difference of award rates among assigned ICs (*Table 1*). Furthermore, it should be noted that applications on these topics were *more likely* to make it past the first hurdle of peer review, that is reaching the point of formal discussion (*Figure 2* and *Table 3*, see line 'Discussed'); thus, if anything, peer reviewers may be slightly biased in favor of AAB-preferred topics.

Our primary analysis focused on first-time submissions of de novo applications in which the award rates were low (<10%). Nonetheless, our analyses of resubmission applications yielded similar findings (Appendix 1). It is also important to note that this was an analysis of applications, not persons. This is an issue when considering multi-PI applications, since all PI's, not just the contact PI, plays a role in choosing the topic of their proposal. Nonetheless, an analysis confined to single PI applications yielded similar findings (Appendix 2).

Only 2% of PIs in this study were African American or Black and thus it is not possible that this small percentage of scientists cover the entire breadth of what NIH funds. The small number of AAB PIs is a source of error for describing AAB scientific interests as reflected in grant applications. Nonetheless, in recent years, NIH has increasingly recognized the importance of enhancing research in targeted topics, as exemplified by a recently announced Common Fund program to enhance transformative research in health disparities and health equity (*NIH, 2021a*) and by the large investment in RADx-UP, a COVID-19 diagnostic testing program in underserved communities (*Tromberg et al., 2020*).

## Conclusion

The lower rate of funding for applications focused on AAB Preferred topics is likely primarily due to their assignment to ICs with lower award rates. These applications have similar peer review outcomes as those focused on other topics (*Table 3*). When AAB preference for each topic (*Equation 1*) was considered as a continuous variable, there was similarly no association between AAB preference and peer review outcomes (*Figure 2*). Topic choice does partially explain race-based funding disparities (*Table 5*, Model 2, absolute value of regression coefficient reduced by 5%), but IC-specific award rates explain the disparities to an even greater degree (*Table 5*, Model 4, absolute value of regression coefficient reduced by 14%). After accounting for IC-specific award rates, we find no association between topic choice and funding outcomes (*Table 6*, Models 2 and 3).

## Additional information

### Funding
No external funding was received for this work.

### Author contributions
Michael S Lauer, Conceptualization, Formal analysis, Supervision, Methodology, Writing - original draft, Project administration; Jamie Doyle, Conceptualization, Formal analysis, Methodology, Writing - review and editing; Joy Wang, Deepshikha Roychowdhury, Formal analysis, Methodology, Writing - review and editing

### Author ORCIDs
Michael S Lauer [iD] https://orcid.org/0000-0002-9217-8177

### Decision letter and Author response
Decision letter https://doi.org/10.7554/eLife.67173.sa1
Author response https://doi.org/10.7554/eLife.67173.sa2

## Additional files

### Supplementary files
- Source data 1. csr_anon_id.RData.
- Source code 1. Main Manuscript.Rmd.
- Source code 2. Appendix 1.Rmd.
- Source code 3. Appendix 2.Rmd.
- Transparent reporting form

### Data availability
The authors have provided the de-identified data frame (in. RData format) along with three R markdown files that will make it possible for interested readers to reproduce the main paper and the two appendices (including all tables, figures, and numbers in the text).

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

## Appendix 1

### Resubmission applications

In our primary analyses, we excluded resubmission applications and competing renewals. Since many applications are not funded on the first try, we repeated our analyses focusing on resubmissions. Our findings are similar.

Of 31,900 resubmission applications received and reviewed, 9605 were funded, for an overall award rate of 30%. There were 448 applications, or 1%, submitted by AAB PIs.

*Appendix 1—figure 1* shows the association of cumulative percentage of AAB applications by the cumulative percentage of topics. There was a nonrandom distribution whereby 11 percent of topics (or 16 topics) accounted for 50 percent of AAB applications. We designted a topic as 'AAB Preferred' if it was among the 16 topics that accounted for 50% of AAB applications.

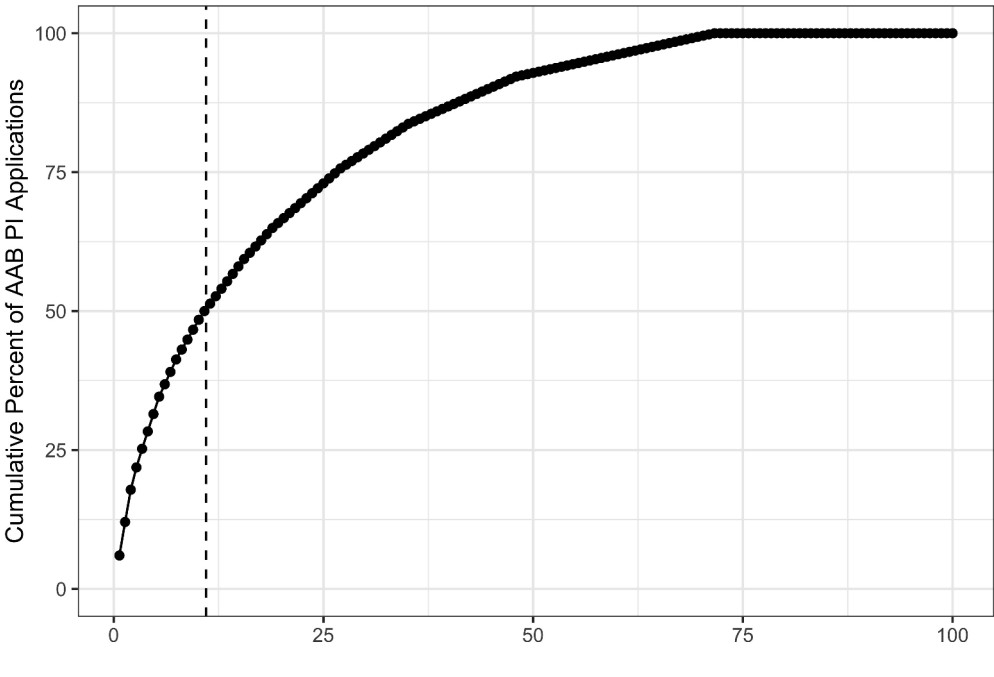

**Appendix 1—figure 1.** Cumulative percentage plot showing the association of cumulative percentage of applications submitted by AAB Principal Investigators and cumulative percentage of topics. Each dot represents a topic; the first dot on the lower left corner shows that this one topic accounted for over 6% of all applications submitted by an AAB PI. Dashed vertical lines show that 11% of the topics accounted for 50% of the applications submitted by AAB Principal Investigators. AAB = African American or Black.

*Appendix 1—figure 2* shows the topic-based peer review outcomes according to the proportion of total AAB applications linked to each topic. There was a non-significant trend whereby applications focusing on topics that accounted for a greater proportion of AAB applications (*Equation 1*) were more likely to make it to discussion (*Equation 5*)(Panel A, p=0.34). When we focused on applications that were discussed and therefore received a priority score, there was no association of mean score for each topic (*Equation 6*) with proportion of total AAB applications linked to each topic (*Equation 1*) (Panel B, p=0.14)

**A**

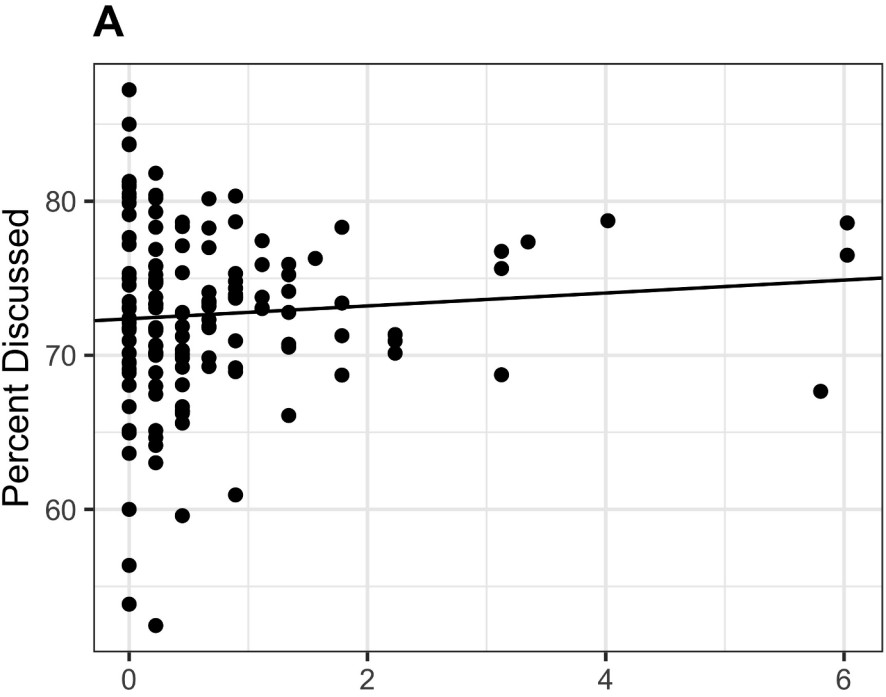

Percent of All AAB Applications

**B**

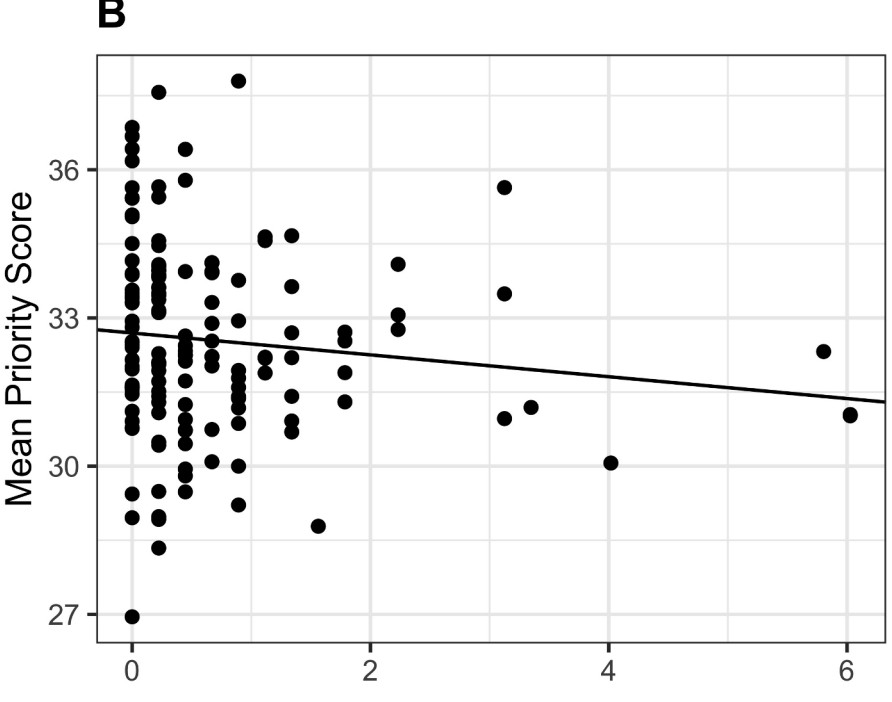

Percent of All AAB Applications

**Appendix 1—figure 2.** Topic peer review outcomes according to proportion of AAB applications linked to specific topics. AAB = African American or Black. Panel **A**: Scatter plot of topic-specific probability of discussion according to proportion of AAB applications. Each dot refers to a topic; the three topics furthest to the right on the X-axis correspond to the three topics at the lower left of

*Appendix 1—figure 2 continued on next page*

*Appendix 1—figure 2 continued*

*Figure 1*. The line is based on a linear regression of proportion of topic-specifc applications discussed on proportion of AAB applications. The slope of the line was not significant (p=0.34). Panel **B**: Scatter plot of topic-specific mean priority score according to proportion of AAB applications. Lower mean priority scores correspond to better, more favorable reviews. Each dot refers to a topic; the three topics furthest to the right on the X-axis correspond to the three topics at the lower left of *Figure 1*. The line is based on a linear regression of proportion of topic-specific mean priority score on proportion of AAB applications; analyses are based on those applications that were discussed. The slope of the line was not significant (p=0.14).

*Appendix 1—table 1* shows peer review and funding outcomes according to whether applications were focused on the 16 topics that made up 50% of all resubmission applications with AAB PIs ('AAB Preferred' topics). Peer review outcomes were similar in the two groups, but applications focusing on AAB Preferred topics were 3% less likely to be funded.

**Appendix 1—table 1.** Application review and funding outcomes according to whether topic was among those that accounted for half of all AAB applications.
Abbreviations as in *Table 2*.

| Characteristic or outcome | | AAB preferred | Other |
|---|---|---|---|
| Total N (%) | | 8193 (25.7) | 23707 (74.3) |
| PI AAB | Yes | 224 (2.7) | 224 (0.9) |
| Discussed | Yes | 6107 (74.5) | 17300 (73.0) |
| Priority Score | Median (IQR) | 31.0 (22.0 to 40.0) | 31.0 (22.0 to 40.0) |
| Percentile Ranking | Median (IQR) | 21.0 (9.8 to 36.0) | 21.0 (10.0 to 35.0) |
| Awarded | Yes | 2419 (29.5) | 7186 (30.3) |

*Appendix 1—table 2* shows probit regressions on topic and funding outcomes. Without consideration of other variables, an AAB preferred topic was associated with a non-significantly lower probability of funding (Model 1, negative regression coefficient for AAB preferred topic). However, after adjusting for IC award rate alone (Model 2, regression coefficient for AAB preferred topic close to zero, p=NS) as well as other IC award rate and other characteristics, there was no association between AAB preferred topics and funding (Model 3, regression coefficient for AAB preferred topic close to zero, p=NS). The IC award rate was strongly associated with likelihood of funding (Models 2 and 3, regression coefficient positive, p<0.001)

**Appendix 1—table 2.** Probit Regression Models (regression coefficients and standard errors) with focus on topic type.
Model 1 shows the univariable association of funding success according to whether the topic is AAB preferred. Model 2 shows results according to topic choice and IC award rate. Model 3 includes early stage investigator status, whether applications had multiple PIs, and whether the application included research on human subjects and/or animal subjects. AIC = Akaike Information Criterion; BIC = Bayesian Information Criterion; Num. obs. = Number of Observations. Other abbreviations same as in *Tables 1* and *2*.

| | Model 1 | Model 2 | Model 3 |
|---|---|---|---|
| Intercept | −0.515*** | −1.397*** | −1.422*** |
| | (0.009) | (0.065) | (0.068) |
| AAB Preferred Topic | −0.023 | −0.001 | −0.004 |
| | (0.017) | (0.017) | (0.018) |
| IC Award Rate | | 0.029*** | 0.029*** |
| | | (0.002) | (0.002) |

*Continued on next page*

*Appendix 1—table 2 continued*

|  | Model 1 | Model 2 | Model 3 |
|---|---|---|---|
| AAB Principal Investigator |  |  | −0.171** |
|  |  |  | (0.065) |
| Early Stage Investigator |  |  | 0.221*** |
|  |  |  | (0.019) |
| Multi-PI Application |  |  | 0.035 |
|  |  |  | (0.020) |
| Human Subjects |  |  | −0.020 |
|  |  |  | (0.018) |
| Animal Subjects |  |  | −0.006 |
|  |  |  | (0.018) |
| AIC | 39034.476 | 38849.763 | 38720.539 |
| BIC | 39051.217 | 38874.874 | 38787.502 |
| Log Likelihood | −19515.238 | −19421.881 | −19352.270 |
| Deviance | 39030.476 | 38843.763 | 38704.539 |
| Num. obs. | 31900 | 31900 | 31900 |

$^{***}p<0.001$; $^{**}p<0.01$; $^{*}p<0.05$.

## Appendix 2

### Single PI applications

In our primary analyses, we included all de novo applications, whether authored by one single PI or more than one PI ('multi-PI'). Here we focus on single-PI applications. Our findings are similar.

Of 76,976 single-PI applications received and reviewed, 6556 were funded, for an overall award rate of 9%. There were 1323 applications, or 2%, submitted by AAB PIs.

*Appendix 2—figure 1* shows the association of cumulative percentage of AAB applications by the cumulative percentage of topics. There was a nonrandom distribution whereby 11% of topics (or 16 topics) accounted for 50% of AAB applications. We designted a topic as 'AAB Preferred' if it was among the 16 topics that accounted for 50% of AAB applications.

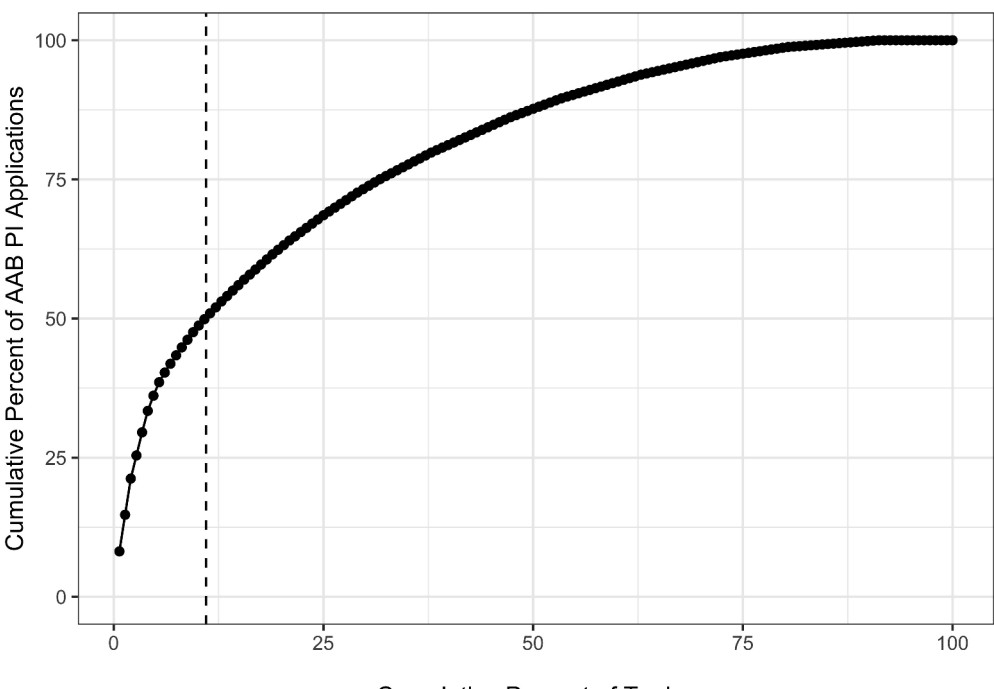

**Appendix 2—figure 1.** Cumulative percentage plot showing the association of cumulative percentage of applications submitted by AAB Principal Investigators and cumulative percentage of topics. Each dot represents a topic; the first dot on the lower left corner shows that this one topic accounted for over 8% of all applications submitted by an AAB PI. Dashed vertical lines show that 11% of the topics accounted for 50% of the applications submitted by AAB Principal Investigators. AAB = African American or Black.

*Appendix 2—figure 2* shows the topic-based peer review outcomes according to the proportion of total AAB applications linked to each topic. There was a non-significant trend whereby applications focusing on topics that accounted for a greater proportion of AAB applications (*Equation 1*) were more likely to make it to discussion (*Equation 5*)(Panel A, p=0.12). When we focused on applications that were discussed and therefore received a priority score, there was no association of mean score for each topic (*Equation 6*) with proportion of total AAB applications linked to each topic (*Equation 1*) (Panel B, p=0.84)

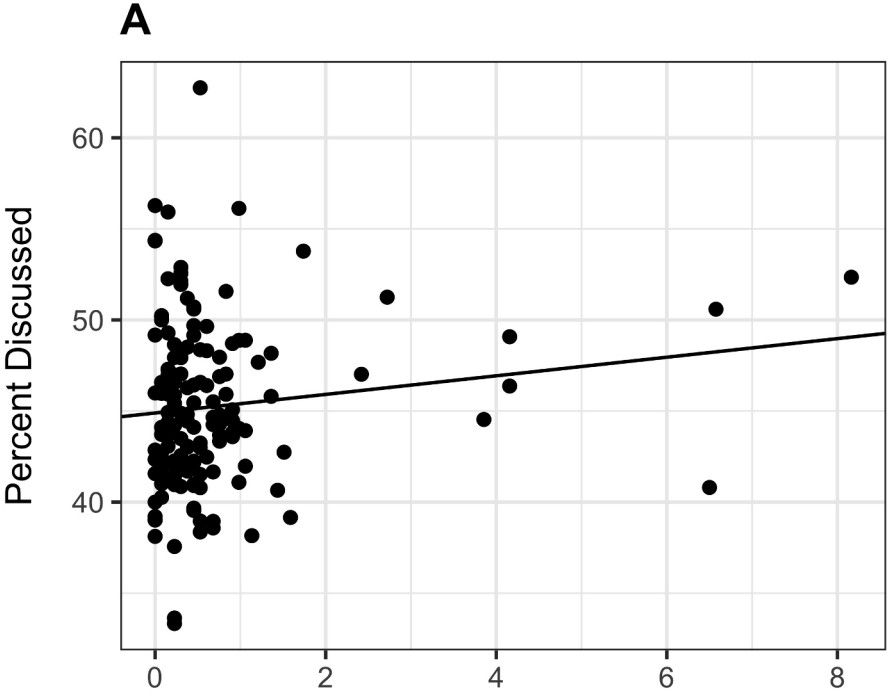

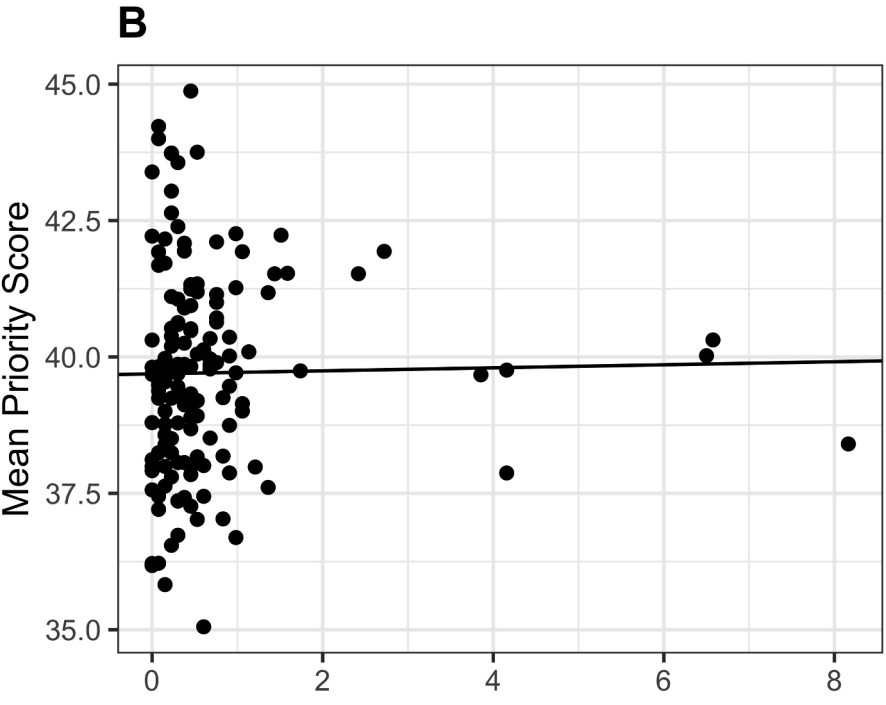

**Appendix 2—figure 2.** Topic peer review outcomes according to proportion of AAB applications linked to specific topics. AAB = African American or Black. Panel **A**: Scatter plot of topic-specific probability of discussion according to proportion of AAB applications. Each dot refers to a topic; the three topics furthest to the right on the X-axis correspond to the three topics at the lower left of

*Appendix 2—figure 2 continued on next page*

*Figure 1*. The line is based on a linear regression of proportion of topic-specifc applications discussed on proportion of AAB applications. The slope of the line was not significant (p=0.12). Panel **B**: Scatter plot of topic-specific mean priority score according to proportion of AAB applications. Lower mean priority scores correspond to better, more favorable reviews. Each dot refers to a topic; the three topics furthest to the right on the X-axis correspond to the three topics at the lower left of *Figure 1*. The line is based on a linear regression of proportion of topic-specific mean priority score on proportion of AAB applications; analyses are based on those applications that were discussed. The slope of the line was not significant (p=0.84).

   *Appendix 2—table 1* shows peer review and funding outcomes according to whether applications were focused on the 16 topics that made up 50% of all resubmission applications with AAB PIs ('AAB Preferred' topics). Peer review outcomes were similar in the two groups, but applications focusing on AAB Preferred topics were 7% less likely to be funded.

**Appendix 2—table 1.** Application review and funding outcomes according to whether topic was among those that accounted for half of all AAB applications.
Abbreviations as in *Table 2*.

| Characteristic or outcome | | AAB preferred | Other |
|---|---|---|---|
| Total N (%) | | 19134 (24.9) | 57842 (75.1) |
| PI AAB | Yes | 660 (3.4) | 663 (1.1) |
| Discussed | Yes | 8993 (47.0) | 26342 (45.5) |
| Priority Score | Median (IQR) | 40.0 (31.0 to 48.0) | 40.0 (30.0 to 48.0) |
| Percentile Ranking | Median (IQR) | 35.0 (22.0 to 45.0) | 34.0 (21.0 to 44.0) |
| Awarded | Yes | 1541 (8.1) | 5015 (8.7) |

   *Appendix 2—table 2* shows probit regressions on topic and funding outcomes. Without consideration of other variables, an AAB preferred topic was associated with a lower probability of funding (Model 1, negative regression coefficient for AAB preferred topic, p<0.01). However, after adjusting for IC award rate alone (Model 2, regression coefficient for AAB preferred topic close to zero, p=NS) as well as other IC award rate and other characteristics, there was no association between AAB preferred topics and funding (Model 3, regression coefficient for AAB preferred topic close to zero, p=NS). The IC award rate was strongly associated with likelihood of funding (Models 2 and 3, regression coefficient positive, p<0.001)

**Appendix 2—table 2.** Probit Regression Models (regression coefficients and standard errors) with focus on topic type.
Model 1 shows the univariable association of funding success according to whether the topic is AAB preferred. Model 2 shows results according to topic choice and IC award rate. Model 3 includes early stage investigator status, whether applications had multiple PIs, and whether the application included research on human subjects and/or animal subjects. AIC = Akaike Information Criterion; BIC = Bayesian Information Criterion; Num. obs. = Number of Observations. Other abbreviations same as in *Tables 1* and *2*.

| | Model 1 | Model 2 | Model 3 |
|---|---|---|---|
| Intercept | −1.361*** | −1.903*** | −1.856*** |
| | (0.007) | (0.023) | (0.030) |
| AAB Preferred Topic | −0.040** | 0.002 | 0.008 |
| | (0.015) | (0.015) | (0.017) |
| IC Award Rate | | 0.060*** | 0.056*** |
| | | (0.002) | (0.002) |

*Continued on next page*

*Appendix 2—table 2 continued*

|  | Model 1 | Model 2 | Model 3 |
|---|---|---|---|
| AAB Principal Investigator |  |  | −0.206*** |
|  |  |  | (0.057) |
| Early Stage Investigator |  |  | 0.160*** |
|  |  |  | (0.016) |
| Human Subjects |  |  | −0.070*** |
|  |  |  | (0.016) |
| Animal Subjects |  |  | −0.038* |
|  |  |  | (0.016) |
| AIC | 44830.312 | 44152.773 | 44026.919 |
| BIC | 44848.814 | 44180.526 | 44091.678 |
| Log Likelihood | −22413.156 | −22073.386 | −22006.460 |
| Deviance | 44826.312 | 44146.773 | 44012.919 |
| Num. obs. | 76976 | 76976 | 76976 |

***$p<0.001$; **$p<0.01$; *$p<0.05$

