## [Decision Letter]

**Acceptance summary:**

This is an important paper that nicely addresses a topical and timely issue relevant to extramural NIH funding. The knowledge gained from your analyses will undoubtedly add to the vigorous discussion about how to move forward. We believe you have revised the paper according to the comments of the reviewers. It was clear from your edits and explanation that some of the comments were outside of the scope of the paper, and we understand that. In the meantime, the other responses were noted to be thoughtful and thorough.

**Decision letter after peer review:**

Thank you for submitting your article "National Institutes of Health Institute and Center Award Rates and Funding Disparities" for consideration by *eLife*. Your article has been reviewed by 4 peer reviewers, including Cliff J Rosen as the Reviewing Editor and Reviewer #2, and the evaluation has been overseen by Mone Zaidi as the Senior Editor. The following individual involved in review of your submission has agreed to reveal their identity: Carlos Isales (Reviewer #1).

In general there was enthusiasm for the topical nature of the subject and the importance of defining inequities in NIH funding for African Americans. However there were issues raised about the analysis, the absence of significant data sets in 50% of the topics, and approach to the analysis. Notwithstanding these potentially resolvable issues, there was consensus that a revised paper will be critical to enhance discussion of this important area and move toward possible solutions.

Essential Revisions

Please refer to individual Recommendations for the Authors, and specifically focus on:

1. Further defending the overall approach to the analysis by providing new data and/or offering explanations to issues raised in the four reviews.

2. Further refine statistical analysis particularly as noted by Reviewer 3.

Reviewer #1 (Recommendations for the authors):

Please respond to the following questions by explaining further and/or providing data.

1. The authors don't discuss the details about how Institute assignments are made. Agreed that this is done by CSR, a number of variables go into this: (a) the π may attempt to direct the application to a specific Institute or Center by including keywords in abstract or title; (b) the π may request a specific Institute or Center.

2. Is part of the issue related to having an Institute named Minority Health, but then have that Institute be relatively underfunded? Would you suggest increased funding for specific Institutes?

3. Not clear if the numbers included are all the R01's received or just the ones received by CSR. Some R01's go directly to Institutes (e.g. RFA's); as do P01's and training grants. If the analysis is for total grants submitted, in sub analysis, were there any differences in AAB funding rates for those applications that went directly to a specific Institute or Center and were of a different topic that the 10% of 148 topics, i.e were there differences in outcomes in π assigned vs CSR assigned Institutes?

4. For resubmitted R01's do 100% go back to the same Institute-Center or what % are reassigned to a different Institute?

5. How was an "Early-Stage investigator Table 5 defined? Did seniority (# of publications, h-index) and funding success correlate with specific Institutes?

6. Do we know if there were differences between number of applications submitted before successful funding for AAB investigators? Or whether AAB investigators were less likely to submit an A1 application after initial triage?

Reviewer #2 (Recommendations for the authors):

Please address the following issues.

1. It is still problematic why topic choice of community engagement or population studies fare worse overall and particularly at an Institute such as AI compared to GM if both have the relatively same proportion of preferred topics, and both have relatively high budgets compared to other institutes. Please provide data if available.

2. Are there other factors imbedded within ICs or their leadership that determine priorities independent of funding scores?

3. Also, are there one or more ICs that drive the correlations between IC funding and preferred topics or PIs? Please provide any data.

4. In addition it should be noted that only 2% of all PIs are AABs so that represents a potential source of error.

5. The data are only up to 2015; it has now been five years and things have changed dramatically at NIH and in society. Do the authors have more recent data? There are now many more multiple π applications including AABs that may not be the contact π yet are likely to choose to be in a preferred topic area.

6. Please discuss potential resolutions to this issue; In other words we now know that the disparity in funding is such that AAB RO1s do worse than white PIs because they are selecting topics that end up at institutes with lower funding rates. Should the institutes devote a set aside for these topic choices to balance the portfolio of the IC and equal the playing field for AABs?

Reviewer #3 (Recommendations for the authors):

1. The limited introductory material, and the Discussion do not appropriately place this important, indeed critical, work in the context it deserves. In general, the expectation for a general science journal such as *eLife* demands greater attention to scholarship and contextualization for the anticipated broad audience. Indeed, the Cover Letter argues that this work is relevant to the "NIH extramural research community" and that it is correcting a "mistaken belief" related to their Hoppe et al. 2019 paper. Furthermore, the authors reference the Taffe and Gilpin *eLife* Feature Article which explains at some length why these matters are of such pressing importance. This is indeed a very important study to publish, although perhaps it should be understood, and written, more as an extension of the ongoing inquiry into NIH funding practices and less as a correction of "mistaken belief" related to the prior paper. It is suggested that at the very least the Introduction and Discussion should address the impact of the funding disparity on (1) AA/B researchers and (2) research on the health concerns of Black Americans, in a scholarly way, including reference to recent materials on this topic to guide the interested reader.

2. Introduction Ln26-41: This section, currently the bulk of the Introduction, fails to describe why applications in some topics would be assigned to one IC over another. The "for example" section is a misdirection and distracting. The key question, given the data in Figure 1 and Table 1, is why a given grant would be assigned to NIMHD or NINR instead of the disease-domain parent institute.

2. The Methods indicate that the 15 topics that accounted for 50% of the AA/B applications were designated as AA/B-preferred for analysis. There is no rationale or justification provided for this choice. Why not the topics that accounted for 66% or 33% or 75%? It would appear critical to provide a strong justification, and some indication of what impact other choices for what is an apparently arbitrary cutoff would have on the central analyses.

3. The Conclusion section presages what will be takeaway message for many readers, i.e. "likely primarily due to". Yet, from this analysis, this conclusion which is likely to be assumed to apply to all AA/B application leaves any disparity of funding for the topics associated with half of the applications with AA/B PIs ignored.

4. The Methods indicate that the definition of AA/B-preferred ICs was the top quartile (6/24 ICs), but there is no rationale or justification provided for this choice. In the absence of a strong justification, it would appear critical to provide some indication of what impact other essentially arbitrary cutoff choices would have on the central analyses. For example, of the five (Fogerty International Center appears to be omitted due to "privacy concerns") ICs with the highest percentage of AA/B applications shown in Table 1, the two lowest are 2.13% (AI) and 2.06% (ES) which are much closer to 12 remaining ICs in AA/B percentage than they are to the third lowest of the AA/B upper quartile. Three of these (DE, 1.99% AA/B; DA, 1.81% AA/B; MH, 1.69% AA/B) are more obviously grouped with AI and ES in terms of AA/B award rate than are the three most extreme AA/B-preferred ICs (HD, NR, MD). This differs from the choices made in Hoppe et al. 2019 where it is fairly obvious from Figure 3A why the 8 topic clusters receiving a high percentage of AA/B applications, and the 8 with zero, were selected for analysis. From Table 1 of this manuscript, the top quartile of ICs account for only 27.3% of AA/B applications. Inclusion of DE, DA and MH would increase this to 48.5% of the AA/B applications, more consistent with the 50% threshold used by the authors to designate AA/B-preferred topics. These choices are not explained or justified and it is critical to give some indication of whether the central conclusions depend on such arbitrary choices.

5. Table 1 is missing a statistic which is key to the authors' conclusions. The award rate for applications with AA/B versus white PIs should be listed per IC, instead of only the aggregate award rate. The entire argument hinges on the presumption that there is no difference in the white/AA/B success rate ratio across the ICs, yet we have seen no analysis of this in prior papers nor does the information occur here.

6. Page 3, Line 102: The phrase "submitted [applications] to" in the context of IC assignment is an improper misrepresentation. This requires correcting throughout, including the Abstract and Line 163 of the Discussion. While applicants can request assignment to a given I or C, this is not always respected and thus this phrasing should reflect NIH assignment, not π choice. This aspect of application assignment (and any dual assignments and selection/refusal to pick up as a secondary IC) should be incorporated into the analysis. Are applications with AA/B PIs more or less likely to be assigned by the NIH's receipt and referral process to the lower-success institutes instead of the parent IC? Are applications with AA/B PIs more likely to request those low-success rate ICs? Are assignment requests equivalently respected? Are applications with AA/B PIs more or less likely to be dual-assigned and/or selected for funding by the secondary IC? What about the role of shared funding between ICs?

7. Page 5, Line 123: While the regression coefficients are useful for a sophisticated audience, it is essential to report the results in a more easily understandable manner for a more general audience. Most importantly the results should be expressed in a way that is comparable with the prior analyses provide by Ginther, Hoppe and their colleagues and indeed is mentioned in the first paragraph of the Introduction. The reader should be informed clearly what percentage of the disparity is accounted for here, similar to the ~20-25% of the racial bias effect accounted for by topic in Hoppe et al. 2019 and by CV metrics in Ginther et al. 2018. This will then help to make the Abstract and Conclusion statements more precise.

8. Figure 1 is confusing in juxtaposition with Table 1 and in context with the overall thrust of the paper. It is suggested that an additional panel be added in which the X axis is the AA/B percentage of applications to that IC, as in Table 1. That is what the reader is expecting at first, and it takes some close reading to figure out these are rankings of the ICs based on submissions by topic. In addition, the six AA/B-preferred ICs should be clearly indicated on-figure by means of colored text, bolded text, a circle or something similar.

9. Discussion, page 10: The second paragraph refers to the funding ecology and ICs with lower award rates, reflecting language used throughout the manuscript. There does not appear to be any description of this ecology and the general science reader will not understand the tremendous disparity in IC funding (e.g., NIMHD and NINR, the extreme outliers in % of AA/B π applications, received 0.8% and 0.4% of the FY2000 budget) without this information. It is essential to add columns in Table 1 that show the average amount of money assigned to each IC across the relevant FYs and to also represent this as a percentage to assist the reader. This context then needs to be mentioned in both Introduction and Discussion where points about IC award rates and funding "ecology" are being made. By way of analogy, the readership of *eLife* expects an extensive mechanistic explanation without key omissions to accompany empirical results and this is similar.

10. Discussion, page 10: The second paragraph is an outline of the illogic captured in this entire investigation. It cleverly draws indirect lines from the overall AA/B disparity of award to the success rate of ICs while seemingly ignoring the fact the analysis reflects only the topics preferred by half of the AA/B applications and reports on only 27.3% of AA/B applications. It is unclear why the primary analysis approach for this inquiry is not simply to examine white vs AA/B success rates across each of the ICs. This would be a much simpler and intuitive answer to what appear to be the authors originating question. The Hoppe report admitted, but tried to undersell, the critically important finding that even in the least-successful quintile of topics, the AA/B applications faired more poorly and it would be critical to understand if that is also true in this analysis of IC assignment.

11. Discussion, page 10: The third and fourth paragraphs are written in imprecise language that does a disservice to the reader understanding of the size of the effects that are being reported. The Hoppe sentence should just include the ~20% number. The comments about "just as well", "even better", "fare a bit worse" and "to a much lower degree" need quantitation.

12. Discussion, page 11, Line 179: The statement is utterly speculative and should be removed or backed with better evidence.

13. Conclusion, page 11: The use of "explain disparities to an even greater degree" needs to be accompanied by an estimate of the effect size, similar to the data in Hoppe underlying the "topic choice does partially explain" beginning of the sentence. In Hoppe, it was something on the order of accounting for only 21% of the white/AAB disparity of grants that were discussed. In this manuscript the authors need to be clearer about what amount of which racial disparity does "an even greater degree" reflect?

14. Conclusion, page 11: The third sentence again inappropriately draws conclusions from the indirect logic available from the design. The authors need to show the IC-specific award rates for white and AA/B applicants within each IC and contrast these by IC-specific award rate if they wish to make this statement.

15. Supplementary Materials: As mentioned above, there is not a rationale provided for excluding the competing continuation applications from the main analysis. This is peculiar given that these were included in Hoppe and the current manuscript is framed as an explanation of those prior results. If the outcome is the same, it would make more sense to make the full sample the subject of the main paper and the more selected sub-sample the supplement.

Reviewer #4 (Recommendations for the authors):

The paper is well written and uses appropriate methods in the described analysis. While this analysis does contribute to understanding the differential outcomes that have been previously described in the funding of AAB investigators, there are still unanswered questions about the success rate differentials. The topics examined only account for 50% of the applications. If there is little or no biases as is suggested by the authors, the other 50% should be able to be accounted for using the analysis performed in this paper. The authors should consider adding a bit of conditional conclusion of these findings.

---

## [Author Response]

Essential Revisions

Before going into the Editors’ and Reviewers’ recommendations in detail, I want to summarize a discussion we had regarding some of Reviewer 3’s comments:

– Reviewer 3 states: “The second paragraph is an outline of the illogic captured in this entire investigation. It cleverly draws indirect lines from the overall AA/B disparity of award to the success rate of ICs while seemingly ignoring the fact the analysis reflects only the topics preferred by half of the AA/B applications and reports on only 27.3% of AA/B applications. It is unclear why the primary analysis approach for this inquiry is not simply to examine white vs AA/B success rates across each of the ICs.”

– We did not focus on white vs AA/B because that is not the question we were attempting to answer.

– In the original Hoppe paper there is a sentence in the discussion which states, “Our analysis shows that all three of the factors that underlie the funding gap—preference for some topics over others, assignment of poorer scores, and decision to discuss an application—revolve around decisions made by reviewers.”

– The point of this analysis is that preference for some topics over others do *not* revolve around decisions made by reviewers. Instead, the funding preferences are due to IC award rates.

– We agreed that we needed to clarify the purpose of our analyses, and we feel have done that in the revised paper.

– You agreed that an analysis focusing on how peer review and IC’s approach topics (not PI’s) would be of interest to eLife. Therefore we decided to revise the paper and submit this revision to you.

Please refer to individual Recommendations for the Authors, and specifically focus on:1. Further defending the overall approach to the analysis by providing new data and/or offering explanations to issues raised in the four reviews.

We have made a number of changes, including more detailed explanations and additional analyses.

2. Further refine statistical analysis particularly as noted by Reviewer 3.

Notwithstanding our discussion above regarding Reviewer 3’s comments, we have refined and extended our analyses in response to a number of comments with which we completely agree:

– New Figure 1 shows in quantitative terms the skewed nature of AAB π preferences. The Figure will help the reader understand why the 50% cut-off was reasonable.

– Reviewer 3 takes issue with our categorical approach (e.g. identifying 10% of topics that account for 50% of AAB π applications). Therefore, we calculated AAB preference for each topic as a continuous variable (equation 1) and show peer review and funding outcomes according to AAB preference in new Figures 2 and 5. These Figures reinforce the absence of an association of peer review outcomes with AAB preference for certain topics.

– We also present an extended descriptive analysis in new Table 4 showing that using a 70% cut-point, instead of 50%, yields similar findings.

– We took steps to better explain our terms and regression analyses. In the Methods section we present a series of equations to explain our terms, and we refer to those equations in other parts of the manuscript. In the legend to Table 5 we explain what the regression coefficients and p-values mean. In the text in the Results and Discussion sections we attempt to “walk the reader through” each of the models presented in the regression table.

Reviewer #1 (Recommendations for the authors):Please respond to the following questions by explaining further and/or providing data.1. The authors don't discuss the details about how Institute assignments are made. Agreed that this is done by CSR, a number of variables go into this: (a) the π may attempt to direct the application to a specific Institute or Center by including keywords in abstract or title; (b) the π may request a specific Institute or Center.

We provide information, as well as a citation, on Institute assignments.

2. Is part of the issue related to having an Institute named Minority Health, but then have that Institute be relatively underfunded? Would you suggest increased funding for specific Institutes?

This is an important question, but it is related to policy and legislative priorities. Therefore, we feel this is not the right venue for NIH staff to publish such a discussion. However, we did add text in the Discussion on how NIH is prioritizing funding for specific topics.

3. Not clear if the numbers included are all the R01's received or just the ones received by CSR. Some R01's go directly to Institutes (e.g. RFA's); as do P01's and training grants. If the analysis is for total grants submitted, in subanalysis, were there any differences in AAB funding rates for those applications that went directly to a specific Institute or Center and were of a different topic that the 10% of 148 topics, i.e were there differences in outcomes in π assigned vs CSR assigned Institutes?

The vast majority (88%) of the applications were reviewed by CSR. Given that only 12% of applications were reviewed outside CSR, we did not conduct stratified analyses.

4. For resubmitted R01's do 100% go back to the same Institute-Center or what % are reassigned to a different Institute?

As a rule, resubmitted applications go back to the same IC.

5. How was an "Early-Stage investigator Table 5 defined? Did seniority (# of publications, h-index) and funding success correlate with specific Institutes?

We provide a definition for Early-Stage Investigator. The second question is about PI’s, not topics, and therefore beyond the scope of this paper.

6. Do we know if there were differences between number of applications submitted before successful funding for AAB investigators? Or whether AAB investigators were less likely to submit an A1 application after initial triage?

This question is about PI’s, not topics, and therefore beyond the scope of this paper. However, it was addressed in the original Hoppe paper.

Reviewer #2 (Recommendations for the authors):Please address the following issues.1. It is still problematic why topic choice of community engagement or population studies fare worse overall and particularly at an Institute such as AI compared to GM if both have the relatively same proportion of preferred topics, and both have relatively high budgets compared to other institutes. Please provide data if available.

Different IC’s do not have the same proportion of preferred topics. Tables 3 and 4 (which is a new table) shows the data. For example, in new Table 4, using a 70% cut-off, AAB preferred topics were disproportionately assigned to the 6 IC’s that received the highest proportion of applications with AAB PIs (25% vs. 14%).

2. Are there other factors imbedded within ICs or their leadership that determine priorities independent of funding scores?

We explain briefly how ICs make funding decisions and cite IC-specific strategic plans.

3. Also, are there one or more ICs that drive the correlations between IC funding and preferred topics or PIs? Please provide any data.

These data are shown in Figure 4, but we can see how this was not clear. We relabeled the X axis as “Proportion of Applications on AAB Preferred Topic (%).” In the Figure legend we added, “In other words, the X-axis value would be 33 percent if one-third of all IC applications focused on AAB Preferred topics, namely those 15 topics that together accounted for 50% of all applications with AAB PIs.” As suggested by Reviewer 3, we highlight the ICs that received the highest proportion of applications with AAB PIs; Table 2 shows that the award rates for these ICs were lower than for all other ICs.

4. In addition it should be noted that only 2% of all PIs are AABs so that represents a potential source of error.

We agree and added text in the Discussion.

5. The data are only up to 2015; it has now been five years and things have changed dramatically at NIH and in society. Do the authors have more recent data? There are now many more multiple π applications including AABs that may not be the contact π yet are likely to choose to be in a preferred topic area.

Since this is a re-analysis of the Hoppe project, we focused on those data. We agree that an analysis of more recent data would be of value, particularly to look for secular trends, but this is beyond the scope of this paper. We agree with the concerns about multi-PI applications; therefore we include a supplemental analysis on single-PI applications.

6. Please discuss potential resolutions to this issue; In other words we now know that the disparity in funding is such that AAB RO1s do worse than white PIs because they are selecting topics that end up at institutes with lower funding rates. Should the institutes devote a set aside for these topic choices to balance the portfolio of the IC and equal the playing field for AABs?

This is an important question, but it is related to policy and legislative priorities. Therefore, we feel this is not the right venue for NIH staff to publish such a discussion. However, we did add text in the Discussion on how NIH is prioritizing funding for specific topics. Furthermore, we hope that the paper communicates the message that the problem is not purported peer review biases against certain topics (Figures 2 and 3, Table 6 models 2 and 3).

This paper is not stating that the entire disparity in NIH R01 funding between AAB and whites is due to the topics they choose. IC funding ecology is one of many factors that contribute to the funding disparity. This has been clarified in the Discussion section.

Reviewer #3 (Recommendations for the authors):1. The limited introductory material, and the Discussion do not appropriately place this important, indeed critical, work in the context it deserves. In general, the expectation for a general science journal such as eLife demands greater attention to scholarship and contextualization for the anticipated broad audience. Indeed, the Cover Letter argues that this work is relevant to the "NIH extramural research community" and that it is correcting a "mistaken belief" related to their Hoppe et al. 2019 paper. Furthermore, the authors reference the Taffe and Gilpin eLife Feature Article which explains at some length why these matters are of such pressing importance. This is indeed a very important study to publish, although perhaps it should be understood, and written, more as an extension of the ongoing inquiry into NIH funding practices and less as a correction of "mistaken belief" related to the prior paper. It is suggested that at the very least the Introduction and Discussion should address the impact of the funding disparity on (1) AA/B researchers and (2) research on the health concerns of Black Americans, in a scholarly way, including reference to recent materials on this topic to guide the interested reader.

As noted previously, we have clarified that this analysis focuses on topics, not PIs.

2. Introduction Ln26-41: This section, currently the bulk of the Introduction, fails to describe why applications in some topics would be assigned to one IC over another. The "for example" section is a misdirection and distracting. The key question, given the data in Figure 1 and Table 1, is why a given grant would be assigned to NIMHD or NINR instead of the disease-domain parent institute.

We provide information, as well as a citation, on Institute assignments.

2. The Methods indicate that the 15 topics that accounted for 50% of the AA/B applications were designated as AA/B-preferred for analysis. There is no rationale or justification provided for this choice. Why not the topics that accounted for 66% or 33% or 75%? It would appear critical to provide a strong justification, and some indication of what impact other choices for what is an apparently arbitrary cutoff would have on the central analyses.

This is an excellent point. We added analyses on AAB Preference as a continuous variable (equation 1, Figures 2 and 5), present a cumulative distribution plot (Figure 1), and test a different cut-point (Figure 1 and Table 4).

3. The Conclusion section presages what will be takeaway message for many readers, i.e. "likely primarily due to". Yet, from this analysis, this conclusion which is likely to be assumed to apply to all AA/B application leaves any disparity of funding for the topics associated with half of the applications with AA/B PIs ignored.

This is an analysis of topics, not AAB PIs. As noted, we considered topic preferences as continuous and categorical variables (equation 1; Figures 1, 2, and 5; new Table 4). We clarify in the re-written conclusion where our interpretations are based (mainly from the probit regressions shown in Tables 5 and 6).

4. The Methods indicate that the definition of AA/B-preferred ICs was the top quartile (6/24 ICs), but there is no rationale or justification provided for this choice. In the absence of a strong justification, it would appear critical to provide some indication of what impact other essentially arbitrary cutoff choices would have on the central analyses. For example, of the five (Fogerty International Center appears to be omitted due to "privacy concerns") ICs with the highest percentage of AA/B applications shown in Table 1, the two lowest are 2.13% (AI) and 2.06% (ES) which are much closer to 12 remaining ICs in AA/B percentage than they are to the third lowest of the AA/B upper quartile. Three of these (DE, 1.99% AA/B; DA, 1.81% AA/B; MH, 1.69% AA/B) are more obviously grouped with AI and ES in terms of AA/B award rate than are the three most extreme AA/B-preferred ICs (HD, NR, MD). This differs from the choices made in Hoppe et al. 2019 where it is fairly obvious from Figure 3A why the 8 topic clusters receiving a high percentage of AA/B applications, and the 8 with zero, were selected for analysis. From Table 1 of this manuscript, the top quartile of ICs account for only 27.3% of AA/B applications. Inclusion of DE, DA and MH would increase this to 48.5% of the AA/B applications, more consistent with the 50% threshold used by the authors to designate AA/B-preferred topics. These choices are not explained or justified and it is critical to give some indication of whether the central conclusions depend on such arbitrary choices.

We agree with the reviewer that we should beyond one categorical cut-off. See comments to Editor; Figures 1, 2, and 5; Table 4.

5. Table 1 is missing a statistic which is key to the authors' conclusions. The award rate for applications with AA/B versus white PIs should be listed per IC, instead of only the aggregate award rate. The entire argument hinges on the presumption that there is no difference in the white/AA/B success rate ratio across the ICs, yet we have seen no analysis of this in prior papers nor does the information occur here.

This paper is an analysis of topics, not PIs.

6. Page 3, Line 102: The phrase "submitted [applications] to" in the context of IC assignment is an improper misrepresentation. This requires correcting throughout, including the Abstract and Line 163 of the Discussion. While applicants can request assignment to a given I or C, this is not always respected and thus this phrasing should reflect NIH assignment, not π choice. This aspect of application assignment (and any dual assignments and selection/refusal to pick up as a secondary IC) should be incorporated into the analysis. Are applications with AA/B PIs more or less likely to be assigned by the NIH's receipt and referral process to the lower-success institutes instead of the parent IC? Are applications with AA/B PIs more likely to request those low-success rate ICs? Are assignment requests equivalently respected? Are applications with AA/B PIs more or less likely to be dual-assigned and/or selected for funding by the secondary IC? What about the role of shared funding between ICs?

We agree, and changed the language to “assigned to”. We address dual assignments.

7. Page 5, Line 123: While the regression coefficients are useful for a sophisticated audience, it is essential to report the results in a more easily understandable manner for a more general audience. Most importantly the results should be expressed in a way that is comparable with the prior analyses provide by Ginther, Hoppe and their colleagues and indeed is mentioned in the first paragraph of the Introduction. The reader should be informed clearly what percentage of the disparity is accounted for here, similar to the ~20-25% of the racial bias effect accounted for by topic in Hoppe et al. 2019 and by CV metrics in Ginther et al. 2018. This will then help to make the Abstract and Conclusion statements more precise.

In the table legends for Tables 5 and 6, equation 7, Results text, and Discussion / Conclusion text, we take steps to be precise in our explanations, including walking the reader through the tables.

8. Figure 1 is confusing in juxtaposition with Table 1 and in context with the overall thrust of the paper. It is suggested that an additional panel be added in which the X axis is the AA/B percentage of applications to that IC, as in Table 1. That is what the reader is expecting at first, and it takes some close reading to figure out these are rankings of the ICs based on submissions by topic. In addition, the six AA/B-preferred ICs should be clearly indicated on-figure by means of colored text, bolded text, a circle or something similar.

The ranking of ICs is by award rate; we added language to the Table legend. We added a color-legend to specific ICs in Figure 4.

9. Discussion, page 10: The second paragraph refers to the funding ecology and ICs with lower award rates, reflecting language used throughout the manuscript. There does not appear to be any description of this ecology and the general science reader will not understand the tremendous disparity in IC funding (e.g., NIMHD and NINR, the extreme outliers in % of AA/B π applications, received 0.8% and 0.4% of the FY2000 budget) without this information. It is essential to add columns in Table 1 that show the average amount of money assigned to each IC across the relevant FYs and to also represent this as a percentage to assist the reader. This context then needs to be mentioned in both Introduction and Discussion where points about IC award rates and funding "ecology" are being made. By way of analogy, the readership of eLife expects an extensive mechanistic explanation without key omissions to accompany empirical results and this is similar.

We added a column in Table 1 on the 2015 appropriations to each IC.

10. Discussion, page 10: The second paragraph is an outline of the illogic captured in this entire investigation. It cleverly draws indirect lines from the overall AA/B disparity of award to the success rate of ICs while seemingly ignoring the fact the analysis reflects only the topics preferred by half of the AA/B applications and reports on only 27.3% of AA/B applications. It is unclear why the primary analysis approach for this inquiry is not simply to examine white vs AA/B success rates across each of the ICs. This would be a much simpler and intuitive answer to what appear to be the authors originating question. The Hoppe report admitted, but tried to undersell, the critically important finding that even in the least-successful quintile of topics, the AA/B applications faired more poorly and it would be critical to understand if that is also true in this analysis of IC assignment.

See above at the beginning of this letter.

11. Discussion, page 10: The third and fourth paragraphs are written in imprecise language that does a disservice to the reader understanding of the size of the effects that are being reported. The Hoppe sentence should just include the ~20% number. The comments about "just as well", "even better", "fare a bit worse" and "to a much lower degree" need quantitation.

In the text descriptions of the probit regression analyses, we provided quantitative terms.

12. Discussion, page 11, Line 179: The statement is utterly speculative and should be removed or backed with better evidence.

We agree; new Figure 2 provides that evidence.

13. Conclusion, page 11: The use of "explain disparities to an even greater degree" needs to be accompanied by an estimate of the effect size, similar to the data in Hoppe underlying the "topic choice does partially explain" beginning of the sentence. In Hoppe, it was something on the order of accounting for only 21% of the white/AAB disparity of grants that were discussed. In this manuscript the authors need to be clearer about what amount of which racial disparity does "an even greater degree" reflect?

We have rewritten the conclusion to be more quantitative and to indicate where in the probit regression models we base our interpretations.

14. Conclusion, page 11: The third sentence again inappropriately draws conclusions from the indirect logic available from the design. The authors need to show the IC-specific award rates for white and AA/B applicants within each IC and contrast these by IC-specific award rate if they wish to make this statement.

See above at the beginning of this letter.

15. Supplementary Materials: As mentioned above, there is not a rationale provided for excluding the competing continuation applications from the main analysis. This is peculiar given that these were included in Hoppe and the current manuscript is framed as an explanation of those prior results. If the outcome is the same, it would make more sense to make the full sample the subject of the main paper and the more selected sub-sample the supplement.

We try to explain better why it is inappropriate to combine first submissions with resubmissions and why to exclude competing renewals. Both resubmissions and competing renewals receive systematically different peer review outcomes.

Reviewer #4 (Recommendations for the authors):The paper is well written and uses appropriate methods in the described analysis. While this analysis does contribute to understanding the differential outcomes that have been previously described in the funding of AAB investigators, there are still unanswered questions about the success rate differentials. The topics examined only account for 50% of the applications. If there is little or no biases as is suggested by the authors, the other 50% should be able to be accounted for using the analysis performed in this paper. The authors should consider adding a bit of conditional conclusion of these findings.

As noted, we added AAB Preference as a continuous variable (equation 1, Figure 2 and 5) and tested a different cut-point (Figure 1, Table 4).